# A Probabilistic Neuro-symbolic Layer for Algebraic Constraint Satisfaction

## Abstract

In safety-critical applications, guaranteeing the satisfaction of constraints over continuous environments is crucial, e.g., an autonomous agent should never crash into obstacles or go off-road. Neural models struggle in the presence of these constraints, especially when they involve intricate algebraic relationships. To address this, we introduce a differentiable probabilistic layer that guarantees the satisfaction of non-convex algebraic constraints over continuous variables. This probabilistic algebraic layer (PAL) can be seamlessly plugged into any neural architecture and trained via maximum likelihood without requiring approximations. PAL defines a distribution over conjunctions and disjunctions of linear inequalities, parameterized by polynomials. This formulation enables efficient and exact renormalization via symbolic integration, which can be amortized across different data points and easily parallelized on a GPU. We showcase PAL and our integration scheme on a number of benchmarks for algebraic constraint integration and on real-world trajectory data.

## 1 INTRODUCTION

In safety-critical applications, a *reliable* AI system should be uncertainty-aware and deliver calibrated *probabilities* over its predictions. At the same time, it should confidently and consistently assign zero probability to certain states of the world if they are invalid, i.e., if they are violating *constraints*. These constraints can encode prior knowledge such as explicit rules [Marconato et al., 2023, 2024, Bortolotti et al., 2024] that can be crucial for the safety of the system and its users. For instance, they can encode that self-driving cars should avoid crashing into obstacles or go off-road [Xu et al., 2020, Giunchiglia et al., 2023].

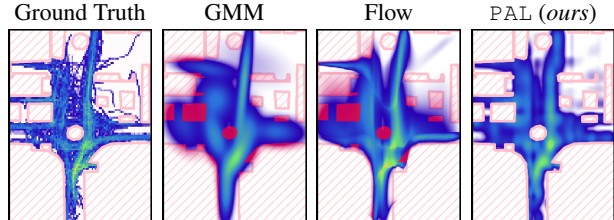

Ground Truth    GMM    Flow    PAL (*ours*)

Figure 1: **PAL is guaranteed to place probability mass only within a given constraint** here represented as the (non deleted) walkable area of a map from the Stanford Drone Dataset, while unconstrained distribution estimators violate the constraint (area shown in red). Details in Section 6.3.

The promise of probabilistic neuro-symbolic (NeSy) methods [Garcez et al., 2019, 2022, De Raedt et al., 2021] for trustworthy AI is to integrate symbolic reasoning over these high-level constraints into neural predictors. The simplest way to achieve this is to encourage constraint satisfaction via a loss penalty at training time [Xu et al., 2018, Fischer et al., 2019, De Smet et al., 2023a]. However, such an approach might have catastrophic consequences at test time, as it does not *guarantee* that invalid configurations will be associated exactly probability zero: even a probability of 0.001% to violate a constraint can be considered harmful in safety-critical applications such as autonomous driving.

A number of recent works overcome this issue by proposing architectures that *certify the satisfaction of given constraints by design* [Manhaeve et al., 2018, Giunchiglia and Lukasiewicz, 2020, Ahmed et al., 2022a, Hoernle et al., 2022]. Unfortunately, they mostly consider constraints over Boolean or discrete variables only [De Smet and Dos Martires, 2024] and extending them to continuous variables is highly non trivial. This is because tackling constraints over *continuous* variables poses unique challenges. First, ensuring modeling a proper distribution over the constraint, i.e., renormalizing a density such that it exactly integrates to 1, is a #P-hard problem [Baldoni et al., 2008] even when the

*Submitted to the 41st Conference on Uncertainty in Artificial Intelligence* (UAI 2025). **To be used for reviewing only**.

distribution and constraints have a relatively simple structure [Zeng et al., 2020b,a]. Second, real-world constraints over continuous variables come in the form of intricate *algebraic* relationships [Barrett et al., 2021, Morettin et al., 2017]. Even considering simple linear inequalities among variables will entail that the support of the induced densities can take the form of (disjunctions of) non-convex polytopes. While focusing on single convex polytope constraints can be easier [Stoian et al., 2024c], it fails to capture real-world scenarios, such as modeling multiple obstacles that need to be avoided simultaneously on a map (Fig. 1). Scalability during learning is another major concern, as one would ideally want to compute fast and exact gradients for each data point. In practice, this is sometimes achieved by approximating or relaxing the constraint [De Smet et al., 2023a,b] but by giving up learning by exact maximum likelihood.

In this work, we narrow these gaps by introducing a *probabilistic algebraic layer* (PAL) that can be seamlessly plugged as the last layer into any neural architecture while guaranteeing probabilistic predictions to satisfy complex algebraic constraints. Our formulation for PAL uses a symbolic integration scheme that scales gracefully as it can be amortized, i.e., computed once for all datapoints, while allowing for exact gradients and hence exact maximum likelihood training, that is being retro-compatible with out-of-the-box optimizers using autograd.

**Contributions.** After formalizing the problem of probabilistic prediction over algebraic constraint in Section 2, we **C1)** introduce PAL and its ingredients while discussing its properties in Section 3; then we **C2)** introduce the GPU-accelerated symbolic polynomial integrator that powers PAL in Section 4. Finally, we **C3)** run an extensive set of experiments, comprising both standard benchmarks for algebraic constraint integration and real-world trajectory data from the Stanford drone dataset [Robicquet et al., 2016], which we augment by manually segmenting constraints representing obstacles and buildings.

## 2 PROBABILISTIC PREDICTIONS UNDER ALGEBRAIC CONSTRAINTS

**Notation**. Uppercase letters denote random variables $(X, Y)$ and lowercase letters denote their assignments $(x, y)$. We use bold for sets of variables $(\mathbf{X}, \mathbf{Y})$, and their joint assignments $(\boldsymbol{x}, \boldsymbol{y})$. We use lowercase Greek letters for denoting algebraic constraints $(\phi)$ and uppercase Greek letters for denoting (vectors of) learnable parameters. We say that $\boldsymbol{y}$ satisfies $\phi$ (written $\boldsymbol{y} \models \phi$) if and only if substituting $\mathbf{Y}$ with $\boldsymbol{y}$ in $\phi$ makes $\phi$ true. We denote the indicator function as $\mathbb{1}\{.\}$, therefore, $\mathbb{1}\{\mathbf{Y} \models \phi\}$ evaluates to 1 for all the values of $\mathbf{Y}$ satisfying $\phi$ and 0 otherwise.

**Setting**. We aim at learning a parameterized conditional distribution of $\mathbf{Y}$ given $\mathbf{X}$, i.e. $p_{\Theta}(\mathbf{Y} \mid \mathbf{X})$, from a dataset

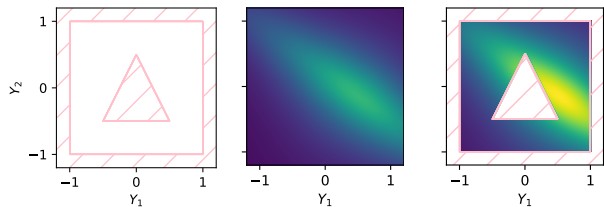

Figure 2: **A renormalized constrained density** (right) obtained by applying **non-convex algebraic constraints** (left) to the support of an unconstrained distribution (middle).

$\mathcal{D} = (\boldsymbol{x}^{(i)}, \boldsymbol{y}^{(i)})$ of realizations of mixed continuous and discrete variables $\mathbf{X}$ and $\mathbf{Y}$. As discussed later, the challenge will be when some $\mathbf{Y}$ are continuous, and we can always assume all of them to be continuous without loss of generality. Differently from a classical supervised learning scenario, the support of $p_{\Theta}$ is not the whole $\mathbb{R}^{|\mathbf{Y}|}$, but a restriction encoded by a constraint $\phi$ defined over $\mathbf{Y}$. This constraint encodes which regions of the label space are invalid, i.e., should have exactly zero-probability and therefore should not be sampled nor predicted [Grivas et al., 2024].

**Which constraints?** We focus on algebraic constraints as Boolean combinations of linear inequalities, which are flexible enough to represent complex supports in the form of disjunctions of non-convex polytopes. We do so in the language of (quantifier-free) satisfiability modulo linear real arithmetic (SMT($\mathcal{LRA}$)) formulas [Barrett et al., 2021]. A constraint in SMT($\mathcal{LRA}$), from here on simply SMT formula or constraint, is a logical formula with arbitrary combinations of the usual Boolean connectives ($\land, \lor, \neg$) over atoms that are restricted to linear inequalities over $\mathbf{Y}$:

$$\left( \sum_i a_i Y_i \bowtie b \right) \quad a_i, b \in \mathbb{Q}, \bowtie \in \{\leq, <, \geq, >, =\}.$$

*Example.* Consider the problem of learning a conditional distribution $p(Y_1, Y_2 \mid X_1, X_2)$ subject to the following constraint $\phi$: $\mathbf{Y} \in [-1, 1]^2 \land \mathbf{Y} \notin \triangle$, where $\triangle$ denotes the triangle with vertices $(-0.5, -0.5)$, $(0, 0.5)$ and $(0.5, -0.5)$, as illustrated in Fig. 2 (left). We can encode such a constraint into SMT($\mathcal{LRA}$) as:

$\phi = (-1 \leq Y_1) \land (Y_1 \leq 1) \land (-1 \leq Y_2) \land (Y_2 \leq 1) \land$
$[(Y_2 < -0.5) \lor (2Y_1 + 0.5 < Y_2) \lor (-2Y_1 + 0.5 < Y_2)]$.

In the more realistic obstacle-avoidance example in the introduction (Fig. 1), $\phi$ can model the surface where cars and pedestrians are allowed to move. One principled solution to design a conditional distribution $p_{\Theta}$ that is constraint-aware is to realize a *product of experts*

$$p_{\Theta}(\mathbf{Y} \mid \mathbf{X} = \boldsymbol{x}) \propto q_{\Theta}(\mathbf{Y} \mid \boldsymbol{x}) \mathbb{1}\{\mathbf{Y} \models \phi\} \qquad (1)$$

where $q_{\Theta}$ is an unconstrained density whose support is the full $\mathbb{R}^{|\mathbf{Y}|}$ (Fig. 2, middle) and $\mathbb{1}\{\mathbf{Y} \models \phi\}$ encodes the satisfaction of the constraint $\phi$. Unfortunately Eq. (1) does not

encode a proper density, as it does not integrate to 1. Alternative ways to train such a product of experts are possible [Hinton, 2002], but they are not retro-compatible with the usual gradient-based optimization recipe to train neural networks: maximum likelihood estimation (MLE). To learn the parameters $\Theta$ by *exact* MLE, and hence to design a layer can be seamlessly plugged-in any network, we would need to compute the renormalization constant of Eq. (1), i.e.,

$$\int q_{\Theta}(\boldsymbol{y} \mid \boldsymbol{x}) \mathbb{1}\{\boldsymbol{y} \models \phi\} \, d\mathbf{Y} \qquad \text{(WMI)}$$

which is also known as a *weighted model integration* (WMI) problem [Belle et al., 2015], i.e., the probability that the constraint is satisfied, $\Pr(\phi)$. Appendix A discusses the background and literature behind WMI in depth.

Here we first note that we treat all variables $\mathbf{Y}$ in Eq. (WMI) to be continuous, something that we can do without loss of generality as we can always reduce a WMI problem over mixed discrete-continuous variables to one over continuous variables only without increasing the problem dimensionality [Zeng and Van den Broeck, 2019]. However, the complexity of solving a WMI problem exactly is #P-hard in general [Zeng et al., 2020a] and tractable algorithms are available only when the constraints $\phi$ come with certain structures [Zeng et al., 2020b]. In the need to scale the computation of Eq. (WMI), many NeSy approaches opted to *approximate* it. For example, DeepSeaProblog (DSP) [De Smet et al., 2023a], arguably the closer system to this work (see Section 5) employs rejection sampling to estimate the WMI integral. Not only does this require to relax the constraints to perform backpropagation, yielding high-variance gradients, but it also hinders scalability, as the WMI integral *needs to be approximated for each datapoint $\boldsymbol{x}$*.

To be able to scale neural networks with complex real-life constraints such as the ones in Fig. 1, while retaining constraint satisfaction guarantees, we have to push the current boundaries of the WMI literature. Next, we discuss how to do so while selecting a parametric form for $q_{\Theta}$ (Section 3) that allows us to efficiently amortize the computation of the WMI integral into a symbolic computational graph that, once built, can leverage GPU parallelism (Section 4).

## 3 A PAL FOR GUARANTEED CONSTRAINT SATISFACTION

We devise a differentiable *probabilistic algebraic layer* (PAL) realizing the following conditional distribution:

$$p_{\Theta}(\mathbf{Y} \mid \boldsymbol{x}) = \frac{q(\mathbf{Y}; \boldsymbol{\lambda} = f_{\boldsymbol{\psi}}(\boldsymbol{x})) \mathbb{1}\{\mathbf{Y} \models \phi\}}{\int q(\boldsymbol{y}'; \boldsymbol{\lambda} = f_{\boldsymbol{\psi}}(\boldsymbol{x})) \mathbb{1}\{\boldsymbol{y}' \models \phi\} \, d\mathbf{Y}'} \qquad \text{(PAL)}$$

where $\Theta = \{\boldsymbol{\lambda}, \boldsymbol{\psi}\}$ and $q(\mathbf{Y}; \boldsymbol{\lambda} = f_{\boldsymbol{\psi}}(\boldsymbol{x}))$ is a flexible unconstrained distribution whose parameters $\boldsymbol{\lambda}$ are the output of a neural backbone $f_{\boldsymbol{\psi}}$ that takes as input $\boldsymbol{x}$ and $\phi$ encodes

an SMT constraint as discussed above. As discussed in Section 2, this construction guarantees that the density $p_{\Theta}$ is non-zero only inside the region defined by $\phi$ (Fig. 2, right). Note that our layer can be used as a standalone distribution estimator when there are no input variables to condition on, i.e., to model $p_{\boldsymbol{\lambda}}(\mathbf{Y})$.

We can design a neural backbone $f_{\boldsymbol{\psi}}$ by easily reusing any existing architecture. Given any (possibly pretrained) neural network $\boldsymbol{z} = h(\boldsymbol{x})$ that outputs an embedding $\boldsymbol{z}$ for each datapoint $\boldsymbol{x}$, in fact we can realize $f$ as $\boldsymbol{\lambda} = g(h(\boldsymbol{x}))$ by adding a simple *gating function* $g$ that maps $\boldsymbol{z}$ to $\boldsymbol{\lambda}$. Less trivial is to select a suitable model family for $q_{\boldsymbol{\lambda}}$ that allows us to efficiently amortize the computation of the denominator of PAL. If we had to deal *only* with Boolean variables $\mathbf{Y}$ and propositional constraints $\phi$, we could leverage probabilistic circuits (PCs) [Darwiche and Marquis, 2002, Choi et al., 2020], compact multilinear polynomials over tractable functions as in Ahmed et al. [2022a]. Unfortunately, there is no equivalent circuit representation for SMT constraints with the required structural properties to guarantee tractability [Vergari et al., 2021].

**General polynomials to the rescue.** Whereas we cannot leverage the properties of structured polynomials such as PCs, we can still employ general (piecewise) polynomials to model the unconstrained density $q_{\boldsymbol{\lambda}}$ in PAL. They will still provide *expressiveness* as they can approximate any density up to arbitrary precision [Morettin et al., 2021, Cheng et al., 2024] and, more crucially, they are closed under integration over a polytope [Zeng et al., 2020b]. The general form of polynomials we consider is therefore:

$$q(\boldsymbol{y}; \boldsymbol{\lambda}) = \sum_{i=1}^{M} \lambda_i \prod_j y_j^{\alpha_{ij}} \qquad (2)$$

where each coefficient $\lambda_i \in \mathbb{R}$ is one of the outputs of the backbone $f_{\boldsymbol{\psi}}$ and the exponents $\alpha_{ij} \in \mathbb{N}$ are constant parameters. As we will discuss in the next section, the complexity of integration will depend, among other things, on the degree of the polynomial and on the number of monomials $M$. At the same time, this will offer the opportunity to design a fixed-parameter tractable integration scheme, as we can bound the polynomial degree by construction.

**How to construct polynomials?** We can generate polynomial structures in an exhaustive way given a max degree $d$, i.e., by generating all possible monomials with degree $\leq d$, or by randomly subsampling them in order to have a compact polynomial of high degree. Alternatively, we can use piecewise polynomials such as *splines*, which have been demonstrated to be very expressive in ML, even with low degree [Durkan et al., 2019b]. In particular, in our experiments, we use cubic Hermite splines [Smith, 1980], see Appendix B.2 for details.

Whereas all the aforementioned polynomials can always be rewritten in the canonical form of Eq. (2), they might not guarantee to model a valid density as they might yield

negative values. To this end, we propose to use *squared polynomials* in PAL, which have been recently investigated in the PC literature for their expressiveness properties [Loconte et al., 2024, 2025b, Loconte and Vergari, 2025]. Specifically, we consider sum of squared polynomials, i.e., polynomials of the form:

$$\sum_k w_k \left( \sum_i u_i \prod_j y_j^{\alpha_{ijk}} \right)^2 \tag{3}$$

where $w_k > 0$ and $u_i \in \mathbb{R}$ and therefore where $\boldsymbol{\lambda} = \{w_k\}_k \cup \{u_i\}_i$ when we rewrite Eq. (3) into Eq. (2). Now we have all the ingredients to discuss how to parallelize the computation of the denominator in PAL.

# 4 EXTREME PARALLELIZATION OF POLYNOMIAL INTEGRALS ON A GPU

State-of-the-art (SoTA) WMI solvers propose various ways to break the WMI integral over $\phi$ into smaller integrals over disjoint convex polytopes $\mu_1, ... \mu_K$ such that $\bigvee_i \mu_i \equiv \phi$. Hence, from here on, we will focus on the problem of integrating a polynomial in canonical form (Eq. (2)) over a single convex polytope. *Our solution to scale the computation of this simpler problem will therefore speed up any SoTA WMI solver*. In practice, for PAL we will adopt SAE4WMI [Spallitta et al., 2024] to break $\phi$ into $\mu_1, ... \mu_K$, as it is the most advanced WMI solver that deals with arbitrary non-convex algebraic constraints $\phi$ at the time of writing. See Appendix A for a discussion.

Our solution, named GPU-Accelerated Simplicial !ntegrator (GASP!), builds upon the idea that we can *compile* once the WMI integral into a highly-parallelizable computational graph $I_\phi(\boldsymbol{\lambda})$ that is a function over the polynomial parameters $\boldsymbol{\lambda}$, and reuse this compiled function to amortize the evaluation of the denominator of PAL for every datapoint $\boldsymbol{x}$. In fact, we can rewrite the WMI integral over a convex polytope $\mu$ as a sum of integrals over monomials:

$$I_\phi(\boldsymbol{\lambda}) = \int q(\boldsymbol{y}; \boldsymbol{\Lambda}) \mathbb{1}\{\boldsymbol{y} \models \mu\} \, d\mathbf{Y}$$
$$= \sum_i \lambda_i \int \prod_j y_j^{\alpha_{i,j}} \mathbb{1}\{\boldsymbol{y} \models \mu\} \, d\mathbf{Y} = \sum_i \lambda_i \eta_i.$$

By treating $\boldsymbol{\lambda}$ as symbolic variables, we have no dependence on $\mathbf{X}$ anymore, and we recover $I_\phi(\boldsymbol{\lambda})$ as a symbolic polynomial whose coefficients are the results of the monomial integrals $\eta_i = \int \prod_j y_j^{\alpha_{i,j}} \mathbb{1}\{\boldsymbol{y} \models \mu\} \, d\mathbf{Y}$. Note that solving all these integrals is an embarrassingly parallelizable problem. While in principle one could use a symbolic solver such as sympy [Meurer et al., 2017a], or another exact WMI solver, these are extremely slow in practice and won't allow PAL to scale, as we empirically confirmed in Section 6.1. We describe this transformation in the Appendix (C.1).

To fully harness GPUs acceleration, we look at ways of further parallelizing the integration of each monomial. An

---

**Algorithm 1** GASP!($q, \mathbf{H}$)

**Input** polynomial $q$ and a convex polytope $\mathbf{H}$
**Output** The Integral $\int_{\mathbf{H}} \hat{q}(\boldsymbol{y}) d\mathbf{Y}$
1: $d \leftarrow$ totalDegree($q$)
2: $(\mathbf{R}, \mathbf{w}) \leftarrow$ prepareGrundmannMöller($d$, dim($\mathbf{H}$))
3: {points and weights of the cubature, see algorithm 4}
4: $\mathbf{V} \leftarrow$ HtoVDescription($\mathbf{H}$)
5: {turns inequalities into vertices spanning $\mathbf{H}$}
6: $\mathbf{S} \leftarrow$ Triangulate($\mathbf{V}$) {obtain simplices}
7: **return** GASPCubature($\hat{q}, (\mathbf{R}, \mathbf{w}), \mathbf{S}$)

---

**Algorithm 2** GASPCubature($q, (\mathbf{R}, \mathbf{w}), \mathbf{S}$)

**Input** a polynomial $q$, cubature points $\mathbf{R} \in \mathbb{R}^{l_{gm} \times n}$ and weights $\mathbf{w} \in \mathbb{R}^{l_{gm}}$, simplices $\mathbf{S} \in \mathbb{R}^{l_s \times (n+1)}$
**Output** the integral $\sum_i \int_{s_i} \hat{q}(\boldsymbol{y}) d\mathbf{Y}$
1: $r_{gasp} \leftarrow$ Array(Length($\mathbf{S}$))
2: **for** $i \leftarrow 1$ **to** Length($\mathbf{S}$) **do** {this loop runs in parallel}
3: $\quad \mathbf{r}_{gm} \leftarrow$ array(length($x$))
4: $\quad vol \leftarrow$ volume($\mathbf{S}[i]$)
5: $\quad$ **for** $j \leftarrow 1$ **to** length($\mathbf{R}$) **do**
6: $\quad\quad$ {this loop runs batched (e.g., 512 points)}
7: $\quad\quad \boldsymbol{x} \leftarrow$ coordinateChange($\mathbf{R}[j], \mathbf{S}[i]$)
8: $\quad\quad$ {transform from unit simplex to $s_i$}
9: $\quad\quad \mathbf{r}_{gm}[j] \leftarrow w[j] \cdot$ polyEval($\hat{q}, \boldsymbol{x}$)
10: $\quad\quad$ {monomials are evaluated in parallel in polyEval}
11: $\quad$ **end for**
12: $\quad \mathbf{r}_{gasp}[i] \leftarrow vol \cdot$ stableSum($\mathbf{r}_{gm}$)
13: **end for**
14: **return** stableSum($\mathbf{r}_{gasp}$)

---

approximate solver such as Sampo [Dos Martires et al., 2019] could leverage the GPU to perform rejection sampling. However, the quality of this approximation scheme will degrade quickly with the number of dimensions and complexity of $\phi$, as observed in Section 6.1.

Our GASP! pushes parallelism to the limit and consists of a further decomposition in sub-problems (Algorithm 1) which in turn are run as small numerical quadrature tasks that can fully leverage the tensor operations of a GPU (Algorithm 2). In a nutshell, we aim to integrate monomials via the Grundmann and Möllers integration formula [Grundmann and Möller, 1978], which is a simple cubature rule that however operates on the unit-simplex.

Therefore, we first decompose each convex polytope $\mu$ into simplices using Delaunay triangulation[1] (L5−7 in Algorithm 1). Then we transform all these simplices in unit-simplices (L5,8 and 12 in Algorithm 2). We finally apply the Grundmann and Möllers cubature rule which reduces to the evaluation of the (transformed) monomials at specific points on the unit-simplex, summing them according

---

[1]We use the QHull library implementation [Barber et al., 1996]

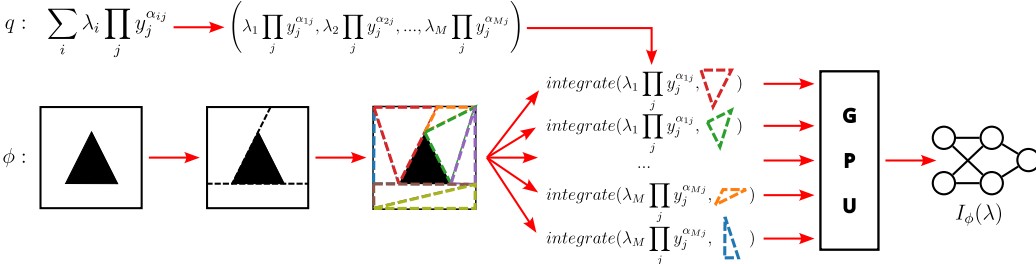

Figure 3: **An overview of our pipeline with `GASP!`.** The integrand $q$ is decomposed into parallelizable monomial integrations (*top*). The non-convex constraint is first decomposed into several convex regions (Appendix A) that are further decomposed into (colored) simplices (*bottom*). The resulting computational graph is highly parallelizable on a GPU.

to some precomputed weights. This cubature rule guarantees exact integration by generating a number of points that is a function of the polynomial degree. Fig. 3 provides an overview of the full integration pipeline in `GASP!`. Note that `GASP!` can be used also as a *standalone* numerical integrator for polynomials, e.g., when there is no neural network $f_\psi$ and the coefficients $\boldsymbol{\lambda}$ are constants.

**Complexity.** Integrating an arbitrary polynomial on a convex polytope is an NP-hard task, which however becomes tractable if the dimensionality is bounded while the polynomial degree and the dimension of the simplex are allowed to vary [Baldoni et al., 2011]. Our scalability will therefore depend on the complexity of the constraint $\phi$ through the number of simplices, on the polynomial degree through the number of monomials and cubature points, and finally on the number of dimensions we integrate over. We provide an in-depth discussion of how these parameters impact `GASP!` in Appendix C.2. We remark that we have to run `GASP!` *only once*, before training, and we can reuse the computational graph $I_\phi(\boldsymbol{\lambda})$ for all datapoints $\boldsymbol{x}$, greatly amortizing computation (see Fig. 4).

**Advanced probabilistic reasoning with `GASP!`** As `GASP!` allows us to feasibly marginalize out all variables $\mathbf{Y}$ in order to compute the WMI integral, it allows us also to compute arbitrary marginals as we integrate out a subset of $\mathbf{Y}$. More crucially, it also allows us to answer other probabilistic reasoning tasks at *test time*, such as exactly computing the probability of satisfying (or violating) a new constraint, e.g., the area around a new obstacle that just appeared on a map like Fig. 1. This can be easily done by computing $\Pr(\phi \wedge \gamma)/\Pr(\phi)$ where $\Pr(\phi)$ is the usual WMI integral, $\phi \wedge \gamma$ the conjunction of the old SMT constraint $\phi$ with a new one $\gamma$ (e.g., representing the obstacle) over which one can compute the updated WMI probability $\Pr(\phi \wedge \gamma)$.

## 5 RELATED WORK

Many probabilistic NeSy [Marra et al., 2024] approaches try to satisfy constraints only *in expectation*. These include incorporating the constraints as a loss term, applied to various formalisms such as propositional logic [Xu et al., 2018, Ahmed et al., 2023], fuzzy logic [Diligenti et al., 2017], physical laws [Stewart and Ermon, 2017] or other algebraic constraints on the outputs of the NNs [Fischer et al., 2019]. Compared to `PAL`, these approaches do not guarantee the satisfaction of the constraints.

A series of other works, instead, guarantees the satisfaction of constraints by *embedding* them in the network architecture. For example, Stoian et al. [2024a,b] take a similar approach to `PAL`, by adding a constraint layer on top of NNs, which however is limited to constraints in the form of conjunctions of inequalities and does provide a probabilistic approach. Multiplexnet [Hoernle et al., 2022] introduces a layer restricted to constraints in disjunctive normal form, which must be relaxed during learning. As a result, it struggles to scale to the intricate constraints in our setting. DeepSade [Goyal et al., 2024] takes a different route, pairing gradient descent with constrained optimization to guarantee SMT constraints for classification and regression tasks. This approach supports SMT formulas with quantifiers, however, its extension to the probabilistic setting is highly non-trivial. Our work is inspired by semantic probabilistic layers (SPL) [Ahmed et al., 2022b], which combines neural networks with probabilistic circuits [Loconte et al., 2025a] in the propositional logic case. We generalize SPL to constraints involving both logical and continuous variables. Similar to SPL, DeepProbLog [Manhaeve et al., 2021] uses probabilistic logic programming to create a layer, but considers only Boolean logic and fully-factorized distributions [van Krieken et al., 2024].

As discussed in Section 2, closer to our work is DeepSeaProbLog (DSP) [De Smet et al., 2023a], which extends DeepProbLog to continuous domains. In contrast to `PAL`, in their implementation, densities are not meant to guarantee constraint satisfaction. In fact, DSP is trained to maximize the probability of the constraint being satisfied, i.e., the WMI integral, via sampling. We detail the DSP-Loss in the appendix. As a side effect, one cannot avoid the rejection layer in DSP as samples directly drawn from the unconstrained probability of a DSP program can violate the constraints. Appendix D discusses additional related works.

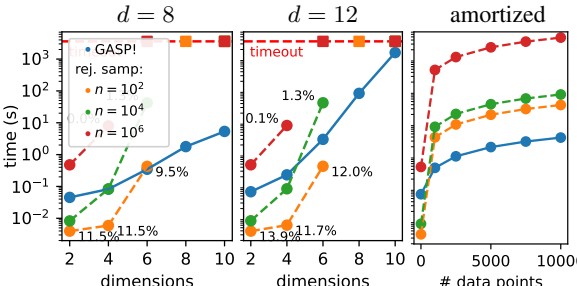

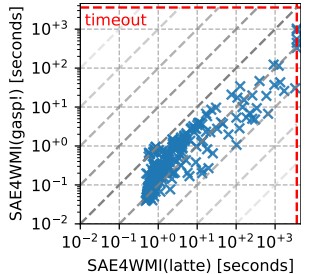

Figure 5: **GASP! can be 1 to 2 orders of magnitude faster than a SoTA polynomial integrator** such as LattE when evaluated on standard WMI benchmarks from Spallitta et al. [2024]

Figure 4: *Left* and *center*: runtime of GASP! and rejection sampling (using $10^{\{2,4,6\}}$ samples) on integrals over 4 random simplices for increasing dimensions. We report the relative error (%) for rejection sampling. *Right*: Our amortization speed-up compared to rejection-sampling in 2-dimensions for the 12-degree polynomial. Fig. 11 shows more results for different degrees and constraints.

## 6 PAL IN ACTION

In this section, we aim to answer the following research questions:[2] **RQ1** How does GASP! compare with other integration solvers? **RQ2** Can PAL learn and scale with an increasing number of constraints? **RQ3** Can PAL handle real-world data and its constraints?

### 6.1 RQ1) BENCHMARKING GASP!

We compare GASP! first against approximate numerical schemes such as rejection sampling, which is commonly used to scale probabilistic NeSy approaches such as DSP (Section 5), and then against SoTA exact polynomial integrators such as LattE [Baldoni et al., 2014].

**GASP! vs numerical approximations for PAL.** We evaluate GASP! w.r.t. an implementation of rejection sampling that runs on the GPU, on random non-convex integration problems of increasing dimensionality and degree. Appendix E.1 details this experimental setting and Fig. 11 reports all results. Here we show in Fig. 4 (*left* and *center*) how rejection sampling struggles to obtain accurate results in higher dimensions for a fixed time budget (1 hr) and polynomials of degree 8 and 12, as the number of rejected samples grow exponentially. Notably, the complexity of the constraints is fixed in this experiment to 4 random simplices. We expect rejection sampling to perform much worse with increasingly complex regions. GASP! instead provides exact computation and scales better (notice that the y-axis is log-scale). More crucially, GASP! can amortize computation as the compiled polynomial $I_\phi(\lambda)$ can be reused throughout the training of PAL. Fig. 4 (*right*) illustrates this for a degree 12 polynomial in 10 dimensions:

---

[2]The source code and the instructions to reproduce our results can be found at: github.com/april-tools/pal

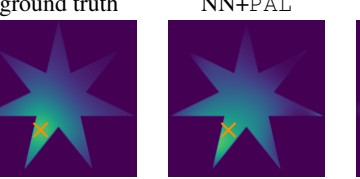

Figure 6: While PAL naturally handles constraints on the $N$-Star, the neural network + GMM has trouble fitting both constraints and data. More quantitative results in Table 1 and qualitative ones in Appendix E.3.1.

integrating over a single convex polytope for $10^4$ datapoints takes less then 10 seconds for GASP! and up to 1 hour with rejection sampling. This is fundamenta to use GASP! in PAL, as the number of evaluations of WMI integrals can easily be orders of magnitude larger than that.

**GASP! as a stand-alone integrator.** We compare GASP! against LattE, a SoTA exact polynomial integrator based on cone decomposition [Baldoni et al., 2014], in the context of WMI integration (details in Appendix E.2). We adopt SAE4WMI (also used in PAL, see Section 4), and compare it while using either GASP! or LattE on 270 WMI instances of varying complexity taken from Spallitta et al. [2024]. Compared to the previous experiment, these WMI problems are highly non-convex, requiring the computation of up to hundreds of integrals each. Notably, here GASP! cannot harness its amortization capabilities and needs to instantiate a different computational graph every time. Despite this, GASP! proves faster than LattE in 263 instances over 270, speeding computation up to one or two orders of magnitude, as shown in Fig. 5. Lastly, as a sanity check we compare GASP! against classical symbolic integrators such as sympy in Appendix E.2.1, achieving a speed up of 1200 times. Overall, our results confirm that GASP! is the best available solver for PAL, positively answering **RQ1**.

### 6.2 RQ2) SCALING CONSTRAINTS WITH PAL

In this experiment, we evaluate PAL in a controlled setting to explore how accurately learning in PAL scales when increasing the number of constraints, while keeping the dimensionality fixed. The task is learning the distribution $p(Y_1, Y_2 \mid X_1, X_2)$ constrained by a $N$-pointed star, as

Table 1: **`PAL` can be more accurate than unconstrained networks and other NeSy baselines** in terms of average test log-likelihood on the $N$Star-dataset. We report the mean over 10 repetitions, more results in Appendix E.3.1.

| | NN + `PAL` | | NN + GMM | | DSP | |
|---|---|---|---|---|---|---|
| $N$ | $d=10$ | $d=14$ | $K=8$ | $K=32$ | by logLike | by Loss |
| 3 | -4.749 | **-4.674** | -4.740 | -4.723 | -5.027 | -42.821 |
| 7 | -4.529 | **-4.527** | -4.708 | -4.612 | -5.019 | -206.411 |
| 11 | **-4.570** | -4.584 | -4.791 | -4.620 | -83.042 | -43.151 |
| 19 | -4.506 | **-4.492** | -4.925 | -4.652 | -5.001 | -155.139 |

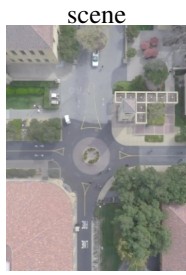 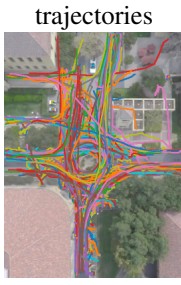 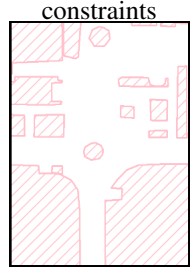

scene  trajectories  constraints

Figure 7: **Our dataset combines challenging constraints with real-world data** on trajectories (*middle*) and aerial maps (*left*) taken from Robicquet et al. [2016]. We manually label the data, indicating invalid areas to move to (*right*).

shown in Fig. 6, for $N = 7$ whose unconstrained distribution is a Cauchy with constant scale and mode $\mathbf{X}$. Table 1 reports the average test log-likelihood of `PAL` versus an unconstrained NN, a mixture density network [Bishop, 1994] with a Gaussian mixture model (GMM) with different components ($K$) as output, and DSP. All models use the same fully-connected NNs with ReLU as a backbone. Appendix E.3.1 further details our setting.

As $N$ increases, the gap between `PAL` and our the competitors widens as the mode tends to be closer to the extremes of the feasible region. Being agnostic to the constraint $\phi$, the NN+GMM has an hard time fitting both the data and respecting the constraints. The same goes for DSP which is hindered by having to fit a single multivariate Gaussian while maximizing the mass inside $\phi$, i.e., $\mathsf{Pr}(\phi)$, see DSP-Loss. This has often the unwanted effect of pushing the learned mode far from the target mode. This aspect prompted us to report the performance of DSP selecting the model according to best log-likelihood on holdout data as opposed to the standard best loss criterion. `PAL` in comparison has an easier time, it naturally handles constraints and therefore just has to move the probability mass around, prompting us to answer **RQ2** positively.

### 6.3 RQ3) STANFORD DRONE DATASET

After showing `PAL` scalability to many constraints, we now evaluate it on real-world data where the ground truth is unavailable. As the majority of existing approaches are unable to process non-convex constraints (Section 5), established benchmarks are lacking. We fill this gap by introducing a new, challenging task based on the Stanford drone dataset (SDD) [Robicquet et al., 2016], which shoes aerial views of pedestrians, cars and bikes traversing different scenes in the Stanford campus. [3] We extract trajectories as in Wu et al. [2023] and we manually annotate two scenes, number 12, which contain the most trajectories, and 2, which contains highly non-convex constraints. Specifically, we create constraints in SMT by segmenting the images over the

---

[3]The dataset can be found at https://github.com/april-tools/constrained-sdd

areas that are non-walkable. Fig. 7 shows an example of an intersection from scene 12 and the extracted constraints. We evaluate two settings: where we model all trajectories together, and when the task is to probabilistically predict the future position given an observed partial trajectory. Differently from RQ1 and RQ2, here we use cubic Hermite splines, as they are easier to train and scale better on real-world data, as detailed in Appendix B.2.

**Modeling joint trajectories ($p(\mathbf{Y})$).** We tackle the problem of estimating the joint distribution of all trajectories, i.e., $p(\mathbf{Y})$. This means we optimize `PAL` as a standalone distribution estimator, without any neural backbone. Appendix E.4 details our experimental setting, in which we compare `PAL` with polynomial splines of increasing complexity against a GMM with increasing number of components and finally a neural spline flow [Durkan et al., 2019a] with multiple transformation layers. `PAL` is able to achieve competitive test log-likelihoods w.r.t. similarly sized GMMs and much larger flows (with one order of magnitude more parameters), as reported in Table 2. More crucially, `PAL` never places probability mass outside the given constraint, while the other competitors do so as shown in Fig. 1 and under $p(\neg\phi)$ in Table 2, denoting the probability of bumping into an obstacle, here approximated with $10^6$ samples for the GMM and the flow. Note that violating the constraint less than 2% of the time can still be greatly harmful for safety-critical applications. In summary, `PAL` is able to guarantee constraint satisfaction while not compromising accuracy, nor time. In fact, `GASP!` takes only 17 seconds on an NVIDIA RTX A6000 to integrate the largest polynomial we consider ($d=12$) on the intricate constraint in Fig. 7 (right).

**Modeling future positions ($p(\mathbf{Y} \mid \mathbf{X})$).** After showing that our layer can effectively model the joint distribution, we consider estimating the distribution over future positions given some observations on the current trajectory. To this end, we subsample five equidistant points from a trajectory that we consider as $\mathbf{X}$ input variables, and we predict the probability of the model of being in any other point $Y_1, Y_2$ in the map. Appendix E.5 completely details our setting.

Table 2: **PAL does not trade expressiveness for constraint-satisfaction** when compared against a GMM and a neural spline flow with $t=1$ and $t=5$ transformations as it provides competitive average test log-likelihood but never violates the constraint for test-set predictions ($\Pr(\neg\phi)$). We report the mean over 10 repetitions, more results in Appendix E.4.

| | | PAL | | GMM | | Flow | |
|---|---|---|---|---|---|---|---|
| | | 10 knots | 16 knots | $K=50$ | $K=100$ | $t=1$ | $t=5$ |
| scene (params) | | (410) | (650) | (300) | (610) | (2670) | (13350) |
| **1** | ll | $-2.98$ | $-2.93$ | $-2.98$ | $\mathbf{-2.91}$ | $-3.10$ | $-2.94$ |
| | $\Pr(\neg\phi)$ | **0.0%** | **0.0%** | $\approx 2.3\%$ | $\approx 1.2\%$ | $\approx 5.6\%$ | $\approx 1.6\%$ |
| **2** | ll | $-3.35$ | $-3.30$ | $-3.34$ | $\mathbf{-3.26}$ | $-3.43$ | $-3.26$ |
| | $\Pr(\neg\phi)$ | **0.0%** | **0.0%** | $\approx 1.2\%$ | $\approx 0.6\%$ | $\approx 2.2\%$ | $\approx 0.7\%$ |

NN+PAL (10 knots)      NN+GMM ($K = 80$)

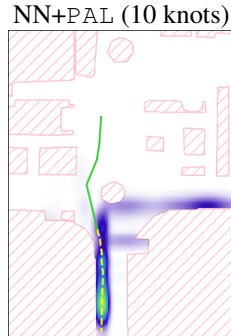 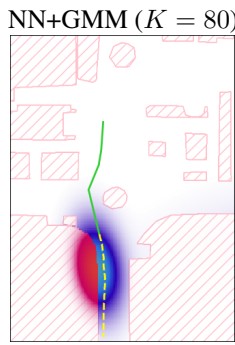

Figure 8: **PAL captures meaningful modes when modelling how a trajectory might evolve ($p(\mathbf{Y} \mid \mathbf{X})$) while never violating the constraint** differently from a NN with a GMM output. The observed test trajectory and ground-truth on scenario 1 completion are reported in green/solid and dashed/yellow. Additional plots in Table 20.

Figure 8 and 9 depict the conditional distributions modeled by PAL and NN+GMM for a single observed trajectory. As expected, multiple future trajectories are visible in the distribution modelled by PAL. In contrast, the conditional GMM is less good at capturing the multimodality of $p(\mathbf{Y} \mid \mathbf{X})$, while also clearly violating the constraints. Table 3 reports a quantitative comparison of NN+PAL with NN+GMM and DSP (finer grained results are reported in Table 19 and 22) showing the average log-likelihood of the points in the (unobserved) trajectory completions and the average probability of sampling future positions that violate $\phi$ (estimated using $10^6$ samples for NN+GMM and DSP). We do not compare against flows in this setting, as parameterized them with a neural network turned out infeasible: it would have required millions of parameters just to realize a linear gating function $g$ (Section 3). Results in Table 3 show that the constraint violation is on average around the 18% for the baselines across all trajectories. This value raises up to 70% for the GMM model on certain trajectories. We highlight that the probability of violating the constraint is higher for conditional predictions because of less data: having direct access

NN+PAL (14 knots)      NN+GMM ($K=32$)

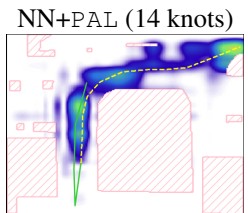 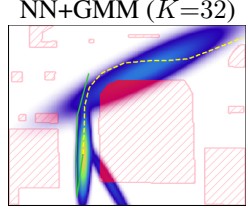

Figure 9: **PAL wraps around the boundaries of our constraints**, in contrast to the NN with an GMM, which predicts a trajectory straight trough the constraint. The observed test trajectory and ground-truth completion on scenario 2 are reported in green/solid and dashed/yellow respectively.

Table 3: **PAL shows competitive likelihoods while guaranteeing constraint-satisfaction** when compared to neural GMM and DSP, for which we provide statistics for the fitted neural distributional fact. We approximate the probability of violating the constraints ($\Pr(\neg\phi)$) at test time numerically per data point and average it. We report the mean over 10 repetitions, further details in Table 19 and Table 22.

| | | NN + PAL | | NN + GMM | | DSP |
|---|---|---|---|---|---|---|
| scene | | 10 knots | 14 knots | $K=50$ | $K=100$ | - |
| **1** | ll | $\mathbf{-2.08}$ | $-2.27$ | $-2.64$ | $-2.83$ | $-3.87$ |
| | $\Pr(\neg\phi)$ | **0%** | **0%** | $\approx 21\%$ | $\approx 20\%$ | $\approx 49\%$ |
| **2** | ll | $-2.23$ | $\mathbf{-2.09}$ | $-2.39$ | $-2.42$ | $-3.61$ |
| | $\Pr(\neg\phi)$ | **0%** | **0%** | $\approx 15\%$ | $\approx 14\%$ | $\approx 36\%$ |

to the constraint $\phi$ greatly improve data efficiency for PAL.

All in all, these results show that PAL, both alone and combined with NNs and in contrast to the other baselines, show promise in effectively modelling complex real world distributions, without trading off expressiveness for constraint satisfaction. We answer **RQ3** positively.

# 7 CONCLUSION

In this work, we introduced PAL, a probabilistic NeSy layer that can be plugged as the prediction layer in any neural network that has to deal with multiple continuous labels and in the presence of algebraic constraints over them. To scale PAL to real-world data, we had to advance the field of WMI by proposing GASP!, a parallelizable polynomial integrator that challenges even established scientific software such as LattE reaching a speed-up of up to 1/2 orders of magnitude. In the future, we plan to investigate and scale GASP! further as a standalone software for a number of computationally intense applications using polynomials such as inference in Bayesian [Zeng and Van den Broeck, 2023] and physics informed neural networks [Lu et al., 2021].

At the same time, PAL offers a number of interesting fu-

ture directions such as enforcing constraints in several applications ranging from fairness [Pfrommer et al., 2022, Ren et al., 2024] to climate modeling [Beucler et al., 2021, Harder et al., 2023, Willard et al., 2020, Beucler et al., 2020] and probabilistic verification [Morettin et al., 2024]. Furthermore, we plan to extend PAL to deal with non-linear constraint while retaining high parallelism [Chin and Sukumar, 2020].

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

# A Probabilistic Neuro-symbolic Layer for Algebraic Constraint Satisfaction (Supplementary Material)

## A BACKGROUND ON WEIGHTED MODEL INTEGRATION

The task of marginalizing over a distribution defined by a density over SMT($\mathcal{LRA}$) constraints is known as *weighted model integration* [Belle et al., 2015]. WMI generalizes weighted model counting (WMC), i.e. the task of summing over the models of a propositional logic formula, to the hybrid logical/continuous domain. In WMC, each satisfying truth assignment $\mu$ is a model, whose weight typically factorizes over the literals:

$$\mathsf{WMC}(\phi, w) = \sum_{\mu \models \phi} \prod_{\ell \in \mu} w(\ell)$$

In WMI, each $\mu$ induces a convex (and disjoint) subregion of $\phi$. For $\mathsf{WMI}(\phi, w)$ to be finite, $\phi$ must encode a closed region. In those cases, each $\mu$ induces a convex polytope containing infinitely many models, i.e. assignments $\boldsymbol{x} \models \mu$. The weight function $w$ can be interpreted in probabilistic terms as an unnormalized density over $\mathbf{X}$. Then, obtaining the weight of each $\mu$ additionally requires integrating $w$ over those models:

$$\mathsf{WMI}(\phi, w) = \underbrace{\sum_{\mu \models \phi}}_{(1)} \underbrace{\int w(\boldsymbol{x}) \mathbb{1}\{\boldsymbol{x} \models \mu\} \ d\mathbf{X}}_{(2)} \tag{4}$$

Computing WMI requires solving two subtasks:

(1) *enumerating* all the $\mu \models \phi$;

(2) *integrating* $w$ inside each $\mu$.

For our purposes, WMI exactly corresponds to the problem of computing the normalizing constant in Eq. PAL:

$$\int w(\boldsymbol{x}) \mathbb{1}\{\boldsymbol{x} \models \phi\} \ d\mathbf{X} = \sum_{\mu \models \phi} \int w(\boldsymbol{x}) \mathbb{1}\{\boldsymbol{x} \models \mu\} \ d\mathbf{X} \tag{5}$$

**Partial enumeration.** In `GASP!`, we solve Eq. 5 by decomposing the integral into convex polytopes first and then further decomposing each polytope into simplices. Clearly, obtaining a compact decomposition of $\phi$ into disjoint convex regions is paramount for reducing the size of the computational graph. In this work, we leverage the enumeration procedure of SAE4WMI [Spallitta et al., 2024] for solving subtask (1). SAE4WMI builds upon a line of work that leverages advanced SMT techniques [Morettin et al., 2017, Spallitta et al., 2022] for minimizing the number of integrations. A key idea of these solver is the enumeration of *partial* (as opposed to *total*) satisfying truth assignments. For the purpose of WMI, the set of satisfying truth assignments $TA(\phi)$ must be complete ($\bigvee_{\mu \in TA(\phi)} \mu \equiv \phi$) and it must contain mutually exclusive/disjoint elements ($\forall i \neq j \ . \ \mu_i \wedge \mu_j \models \bot$). As opposed to total assignments, partial assignments are not required to map every inequality of $\phi$ to $\{True, False\}$.

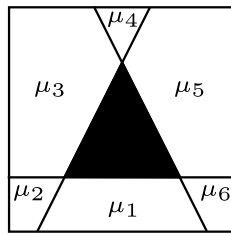 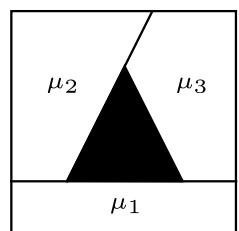

Figure 10: A decomposition of the non-convex constraint in our running example with total (*left*) vs. partial (*right*) satisfying truth assignments.

It is easy to show that the latter can exponentially reduce the number of partitions of $\phi$ by considering a disjunction among $N$ atomic formulas: $\phi = \bigvee_{i=1}^{N} A_i$. While the size of the set of total satisfying truth assignment is $2^N - 1$, we can fully characterize the formula with $N$ disjoint partial assignments $\phi = \bigvee_{i=1}^{N} \mu_i$, where:

$$\mu_1 = A_1$$
$$\mu_2 = \neg A_1 \wedge A_2$$
$$\mu_3 = \neg A_1 \wedge \neg A_2 \wedge A_3$$
$$...$$
$$\mu_N = \neg A_1 \wedge \neg A_2 \wedge ... \wedge \neg A_{N-1} \wedge A_N$$

The benefits of enumerating partial truth assignments is evident even in our lower dimensional example, as depicted in Figure 10.

# B TRAINING AND PARAMETRIZING POLYNOMIALS

We tested `PAL` with two kinds of polynomial density: single polynomials and cubic, hermite splines.

As expressiveness is tied to increasing the total degree, the former might be difficult to employ in fitting very complex distributions. We turned to the latter in the Stanford Drone use cases, partitioning the overall density into equally-sized bins, each encoded as the product of squared univariate splines. Furthermore, we consider a mixture of the parametric form above. The resulting parametric form is both stable to train and expressive, while also leveraging the amortization capabilities of `GASP!` for each bin.

We will first specify the exact re-parametrization and methods we used for both raw polynomials and splines, and then detail the loss.

## B.1 POLYNOMIALS

We experienced some issue during training when trying to directly predict coefficients of single polynomials, due to the different scale-sensitivity of the monomials. We tackled this issue for the $N$-pointed star experiments in by adding a re-parametrization layer in front of the polynomial. Let $I(\mathbf{\Lambda}) = \sum_i \sum_j \mathbf{\Lambda}_i \mathbf{\Lambda}_j \eta_{ij}$ be the integrated, squared polynomial. Then our re-parametrization is the following:

$$r(z_i) = \texttt{sign}(z_i) \frac{\sqrt{z_i + d'} - \sqrt{d'}}{\sqrt{\eta_{ii}}}$$

with $d' = 0.1$. This dampens the impact of coefficients of high-degree monomials, as they are often associated with large $\eta_{ii}$.

Additionally, before training, we initialize the magnitude of the output of our last layer by fitting a constant scalar per dimension using L-BFGS [Liu and Nocedal, 1989] over the first 1000 training samples.

While, with these methods, we are able to train `PAL` for a single, high-degree polynomial in a stable manner, we want to stress that they are not needed for the spline-parametrization.

## B.2 POLYNOMIAL SPLINES

We use univariate, cubic, Hermite splines [Smith, 1980], which we then square to guarantee non-negativity and multiply to create multivariate splines.

Each spline is specified by the value and derivative at the knots. In order to evaluate and integrate, we create the explicit polynomial. We will now focus on a specific bin $[k_i, k_{i+1}]$. We view the spline just as a re-parametrization of a polynomial and compute the coefficients explicitly, in order to plug into the our framework to quickly compute the integral $I(\boldsymbol{\lambda})$. In comparison to the usual way to construct splines, in which the spline is constructed on the standard-bin $[0, 1]$, we have to account for scale. We therefore construct the parameters corresponding to the polynomial on the $[0, k_{i+1} - k_i]$ and shift the monomials $m$ via $m'(x) = m(x - k_{i+1})$ before handing them to GASP!. The flexibility of shifting via a coordinate transform, instead of transforming the parameters of the polynomial, which is numerically instable, shows the adaptability and versatility of GASP!. The computed parameters of GASP! belong to the polynomial on $[0, k_{i+1} - k_i]$ per bin, and are numerically significantly more stable to evaluate than explicitly constructing the parameters of the shifted polynomial on $[k_i, k_{i+1}]$.

Given $k_i$ and $k_{i+1}$, we compute the parameters as follows, which is a straightforward adaptation of the standard construction on $[0, 1]$.

First, we construct the parameters $a + b \cdot x + c \cdot x^2 + d \cdot x^3$ on $[0, 1]$. With $v_i, v_{i+1}$ we denote the value and $v'_i, v'_{i+1}$ the derivative:

$$
\begin{aligned}
a &= v_i \\
b &= v'_i \\
c &= 3 \cdot (v_{i+1} - v_i) - 2 \cdot v'_i - v'_{i+1} \\
d &= 2 \cdot (v_i - v_{i+1}) + v'_i + v'_{i+1}
\end{aligned}
$$

We then scale the polynomial by transforming the parameters. Let $\Delta = (k_{i+1} - k_i)$, then:

$$
\begin{aligned}
a' &= a \\
b' &= b \cdot (1/\Delta) \\
c' &= c \cdot (1/\Delta^2) \\
d' &= c \cdot (1/\Delta^3)
\end{aligned}
$$

These are the coefficients corresponding to the unsquared polynomial. The squared polynomial (of degree $6$) is then just the combination of these parameters with itself, just as the multiplication of two splines is a combination of the respective parameters. Additionally, we take a mixture of these splines. These form $\boldsymbol{\lambda}$.

We integrate by enumerating the bins, shifting the monomials, and then obtain the coefficients for $I(\boldsymbol{\lambda})$ and during training, we view the piecewise spline as a re-parametrization layer that transforms the output of the neural network, so value and derivative at the knots, into the coefficients of the polynomial.

## B.3 LOSS

For PAL, we minimize the following loss:

$$
l(\boldsymbol{x}^{[1:b]}, \boldsymbol{y}^{[1:b]}) = (-1) * \underbrace{\sum_b \log p_{\boldsymbol{\Theta}}(\boldsymbol{y}^{(b)} \mid \boldsymbol{x}^{(b)})}_{\text{constrained log-likelihood}} + \underbrace{\sum_b \mathbb{1}\{\log i_b \geq 10\} (\log i_b - 10)^2}_{\text{penalty on too large values of } I(\boldsymbol{\lambda} = f_\psi(\boldsymbol{x}^{(b)}))} \tag{6}
$$

with $i_b = I(\boldsymbol{\lambda} = f_\psi(\boldsymbol{x}^{(b)}))$. This loss biases the neural network towards more numerically-stable range of integral-values. Due to the scale-invariance, this does not influence the expressivity and only influences numerical stability. Although we use

it for both raw polynomials and splines, it's main use is to stabilize training of raw polynomials as the splines are inherently easier to train.

## C  `GASP!`

### C.1  HANDLING THE SYMBOLIC POLYNOMIAL

We start with an symbolic polynomial $q(\boldsymbol{y}, \boldsymbol{\lambda})$ of the form:

$$q(\boldsymbol{y}, \boldsymbol{\lambda}) = \sum_i \lambda_i \prod_j y_j^{\alpha_{ij}}$$
$$= \sum_i \lambda_i m_i(\boldsymbol{y})$$

where $m_i(\boldsymbol{y})$ denotes the $i$th monomial. This induces the vector-valued version polynomial:

$$\vec{v}_q(\boldsymbol{y}) = \begin{pmatrix} m_{i_1}(\boldsymbol{y}) \\ m_{i_2}(\boldsymbol{y}) \\ \vdots \\ m_{i_n}(\boldsymbol{y}) \end{pmatrix}$$

This function $\vec{v}_q$ is completely independent of $\boldsymbol{\lambda}$, and can be directly plugged into `GASP!` to obtain the integral over each monomial $m_i(\boldsymbol{y})$. This construction also makes it straightforward to parallelize using existing frameworks like PyTorch [Paszke et al., 2019b].

We therefore use the following algorithm for the symbolic integration using `GASP!`:

---
**Algorithm 3** SymbolicIntegral($q$, $\mathbf{H}$)

---
**Input** Polynomial $q(\boldsymbol{y}, \boldsymbol{\lambda})$, Polytope $\mathbf{H}$
**Output** $I(\boldsymbol{\lambda}) = \int_{\mathbf{H}} q(\boldsymbol{y}, \boldsymbol{\lambda}) d\mathbf{Y} = \sum \lambda_i \eta_i$
  1: $\vec{v}_q \leftarrow$ ToVectorValued($q$)
  2: $\eta \leftarrow$ GASP!($\vec{v}_q$, $\mathbf{H}$) {see Algorithm 1}
  3: **return** $\boldsymbol{\lambda} \rightarrow$ I($\boldsymbol{\lambda}; \eta$)

---

### C.2  DETAILED ANALYSIS OF `GASP!`

We will now detail the algorithm used to solve the non-symbolic integration problem over the convex polytope $\mathbf{H}$: $\int_{\mathbf{H}} q(\boldsymbol{y}) d\mathbf{Y}$, where $q$ denotes a polynomial that is only over $\boldsymbol{y}$ (not $\boldsymbol{y}$ and $\boldsymbol{\lambda}$).

The main entry-point for `GASP!` is Algorithm 1, which takes a polynomial $q$ and a convex polytope $\mathbf{H}$ in $\mathcal{H}$-description, so defined via $\mathbf{H} = \{\boldsymbol{y}|\mathbf{A}\boldsymbol{y} \leq \mathbf{b}\}$, and returns the integral $\int_{\mathbf{H}} q(\boldsymbol{y}) d\mathbf{Y}$. Being based on a cubature integration-formula over the unit-simplex, the first step is first querying the total degree of $q$ and then creating the cubature points and weight (L2, Algorithm 1). The total number of points and weights depend on the degree, but are exact for any polynomial up to the respective degree. The cubature points and weights follow theorem 4 of Grundmann and Möller [1978], also provided in Algorithm 4. Given a polynomial of total degree $d$ and dimension $n$, enumerating the points has a complexity of $\mathcal{O}(r \cdot \binom{r+n-1}{n-1})$ for $r = \lceil \frac{d}{2} - 1 \rceil$. In practice, this is done on the CPU and re-used for every polytope we want to integrate over. We then move our polytope $\mathbf{H} = \{\boldsymbol{x} \mid \mathbf{A}\boldsymbol{x} \leq \mathbf{b}\}$ from $\mathcal{H}$ description into its $\mathcal{V}$-description and call it $\mathbf{V}$ (L4, Algorithm 1). This operation has a complexity of $\mathcal{O}(m^{\lfloor n/2 \rfloor})$, with $m$ being the number of inequalities [Chazelle, 1993], so polynomial for some fixed dimension $n$. Afterwards, we triangulate the vertices $\mathbf{V}$ into an array of simplices $\mathbf{S}$ (L6, Algorithm 1). While finding the minimal triangulation is NP-complete [Kaibel and Pfetsch, 2002], finding some triangulation using the Delaunay-algorithm can be done in $\mathcal{O}(v^{\lceil n/2 \rceil})$ [Amenta et al., 2007], with $v$ being the number of vertices obtained

previously. These computations are also executed on the CPU using QHull [Barber et al., 1996]. We are now prepared for the actual numerical integration, which will happen on the GPU (L7, alg. 1).

This algorithm is detailed in Algorithm 2. It takes the polynomial $q$, points and weights $(\mathbf{R}, \mathbf{w})$ from the cubature rule and simplices $\mathbf{S}$. In practice, these are PyTorch [Paszke et al., 2019a] tensors. We then loop over each simplex in $\mathbf{s}_i$ (L2, Algorithm 2) and compute the cubature over all cubature points and weights $(\mathbf{R}, \mathbf{w})$ (L6, Algorithm 2). As these cubature points are distributed over the unit-simplex, we need a coordinate change to transform the points from the unit-simplex to points on $\mathbf{s}_i$ (L8, Algorithm 2), which is just a matrix-vector multiplication. We also need to calculate the absolute determinant of the Jacobian of this transformation for the pullback-measure, which coincides with the volume of the simplex (L5,Algorithm 2) and has the complexity $n^3$ due to the determinant. We then evaluate the polynomial on each point and add the sum weighted according to the cubature weight $\mathbf{w}$ (L10 and 13, Algorithm 2). In the end we sum up our integral over each simplex to arrive at the integral over our polytope (L15, Algorithm 2). In order to increase numerical accuracy, we first divide into positive and negative parts, sort each, and then sum up the elements via stableSum. In practice, this is done per batch in the inner loop, as it runs batched. Every loop in this algorithm, including over multiple monomials arising due to the symbolic integral, is done in parallel and leverages the parallelism of the GPU.

Finally, we want to stress that every computation in our algorithm uses established, heavily optimized numerical routines. This approach allows us to exploit the unique performance characteristics and heavy parallelism of GPUs to tackles this challenging problem. Therefore, the overall complexity of algorithm 2 is $\mathcal{O}(l_s \cdot (l_{gm} \cdot (n^2 + \log l_{gm}) + n^3) \log l_s)$, with $l_s$ being the number of simplices and $l_{gm}$ the number of cubature points.

This enables us to compute the overall complexity of gasp by noting that the number of simplices grows with $\mathcal{O}(m^{\lceil n/2 \rceil^2})$ [Seidel, 1995] with $m$ being the number of inequalities. We arrive at $\mathcal{O}(m^{\lceil n/2 \rceil^2} \cdot (l_{gm} \cdot (n^2 + \log l_{gm}) + n^3) \lceil n/2 \rceil^2 \log m)$ and $l_{gm} = r \cdot \binom{r+n-1}{n-1}$. While this complexity seems unwieldy, we first want to note that the problem is fundamentally hard, as integrating an arbitrary polynomial over a single simplex is already NP-Hard [Baldoni et al., 2011]. Furthermore, in many applications, we can assume $dim(\mathbf{Y}) \ll dim(\mathbf{X})$ and therefore $dim(\mathbf{Y})$ is actually reasonable. Finally, we can amortize this computation as it must only be done once per constraint, enabling fast and efficient training.

### C.2.1 Grundmann-Möller-Cubature

---
**Algorithm 4** PrepareGrundmannMöller$(d, n)$

---
**Input** Total degree $d$, dimensions $n$
**Output** Cubature points $\mathbf{R} \in \mathbb{R}^{l_{gm} \times n}$ and weights $\mathbf{w} \in \mathbb{R}^{l_{gm}}$

1:  $\mathbf{R} \leftarrow []$
2:  $\mathbf{w} \leftarrow []$
3:  $s \leftarrow \lceil \frac{d}{2} - 1 \rceil$
4:  {According to Theorem 4 of Grundmann and Möller [1978]}
5:  **for** $i \leftarrow 0$ **to** $s$ **do**
6:     $w_i \leftarrow (-1)^i 2^{-2s} \frac{(d+n-2i)^d}{i!(d+n-i)!}$
7:     $\Gamma \leftarrow$ combinationsSummingTo$(n, s - i)$
8:     {All combinations of $n$ natural numbers summing to $s - i$}
9:     **for** $\gamma \in \Gamma$ **do**
10:      $\mathbf{r} \leftarrow \left( \frac{2\gamma_0 + 1}{d+n-2i}, \dots, \frac{2\gamma_n + 1}{d+n-2i} \right)$
11:      Append$(\mathbf{R}, \mathbf{r})$
12:      Append$(\mathbf{w}, w_i / \text{Len}(\Gamma))$
13:    **end for**
14: **end for**
15: **return** $\mathbf{R}, \mathbf{w}$

---

## D ADDITIONAL RELATED WORKS

**Other constraints in deep learning.** Many approaches have been proposed for dealing with a variety of specialized constraints in neural networks. An abundant body of work investigates architectures that guarantee specific properties, such

as monotonicity [You et al., 2017] or permutation invariance of inputs [Zaheer et al., 2017]. Other works impose constraints over they dynamics of a neural network, e.g., to follow a physics-based constraint or a PDE [Raissi et al., 2019, Li et al., 2020, Beltran et al., 2024]. These approaches are orthogonal to ours and cannot be easily generalized in our framework where constraints are algebraic.

**Sampling with constraints.** It is well known that (algebraic) constraints pose significant challenges for sampling procedures. Afshar et al. [2016] addressed these challenges with Markov Chain Monte Carlo, which is however unsuited for training `PAL` via gradient descent. Abboud et al. [2022] introduce an FPRAS scheme to approximate WMI problems whose constraints are represented in disjunctive normal form (DNF). While this approach provides some guarantees, it would still require many samples to get an accurate estimate of a WMI integral and DNFs are not compact representations of many real-world constraints [Hoernle et al., 2022]. Another recent work on sampling under algebraic constraints is the Disjunctive Refinement Layer [Stoian and Giunchiglia, 2025], which is an iterative projection onto non-convex sets defined by quantifier-free conjunctions, disjunctions and negations of linear inequalities. It allows for sampling with guaranteed constraint satisfaction, but the iterative projection of the invalid probability mass leads to a clustering of projected samples at the boundaries and obstructs gradient flow.

**Learning constraints.** The problem of learning SMT($\mathcal{LRA}$) constraints from positive/negative examples was first addressed by INCAL [Kolb et al., 2018b], an incremental approach built upon SMT solvers. LARIAT [Morettin et al., 2020] addressed the problem of jointly learning the SMT($\mathcal{LRA}$) constraints and a piecewise polynomial density from unlabelled data. These two components are learned separately and then combined into a probabilistic model that can be queried using WMI solvers. In this paper, we assume the constraints are given and they are accounted for when learning the parameters of the density function. Combining `PAL` with the above approaches is an interesting future direction.

# E EXPERIMENTAL DETAILS

## E.1 REJECTION SAMPLING VS. `GASP`!

The polynomials with dimension $n$ and total degree $d$ we want to integrate are all of the form

$$\left( \sum_{\substack{\boldsymbol{\alpha}_i \in \mathbb{N}^n \\ 1^T \boldsymbol{\alpha}_i \leq d}} c_i \prod_j y_j^{\alpha_{ij}} \right)^2 .$$

The coefficients $c_i$ are distributed as follows: $c_i \sum \pm \text{Poisson}(2)$, where $\pm$ denotes a random, equal chance, sign.

We generate the random simplices by following the same procedure as many times as needed:

1. draw a random unit simplex;
2. scale it between $0.5$ and $1.5$ (uniformly);
3. transform the vertices using a random orthonormal matrix;
4. translate it between $0$ and $6$;
5. keep the simplex if it does not overlap any previous simplex.

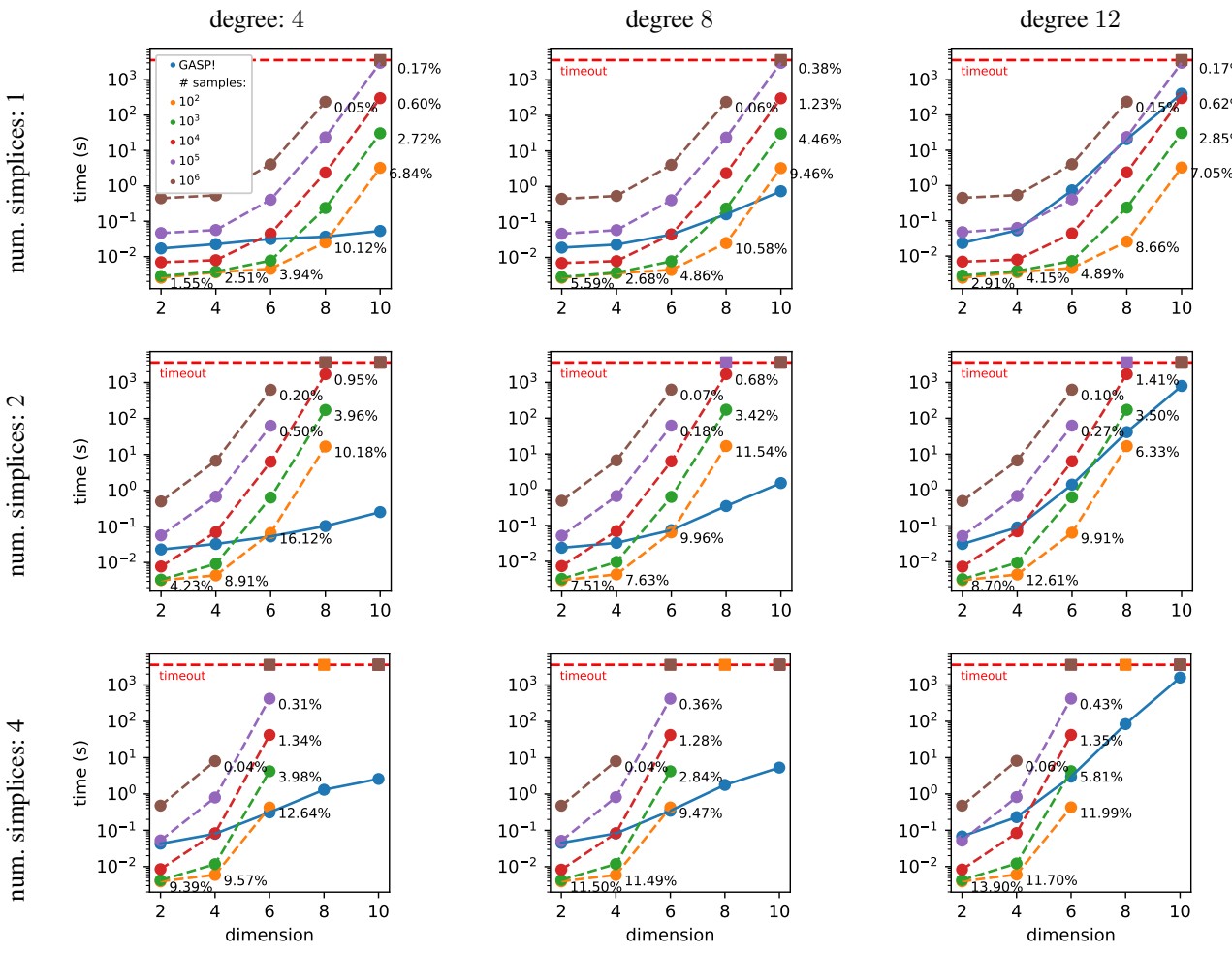

Figure 11: Runtime (in seconds) of rejection sampling vs a single GASP!-run for integrating random polynomials of varying degree over 4 random simplices. For rejection sampling, we additionally report its relative error. This benchmark was run using an NVIDIA RTX A6000, an AMD EPYC 7452 32-Core Processor and 528 Gigabyte RAM.

## E.2 SAE4WMI(`LATTE`) VS. SAE4WMI(`GASP!`)

For this experiment, we used the benchmarking suite originally released with SAE4WMI [Spallitta et al., 2022] (https://github.com/unitn-sml/wmi-benchmarks) for generating random WMI problems. These instances are composed of a SMT($\mathcal{LRA}$) formula encoding the support of the distribution and a piecewise polynomial weight function. The weight functions have arbitrary SMT($\mathcal{LRA}$) conditions as internal nodes and non-negative polynomial leaves. In contrast with the benchmarks employed by Spallitta et al., our weight functions do not have sums or product as internal nodes. This minor modification was made to enforce a tighter control over the maximum overall degree of the weight function. The problems have $n \in \{3, 4, 5\}$ real variables and are generated in a recursive manner, with formulas and weight functions having depth $r \in \{2, 3, 4\}$, the latter having polynomial leaves with maximum degree $d \in \{0, 2, 4\}$. For each configuration of $\langle n, d, r \rangle$, we generated 10 instances, for a total of 270 WMI problems. This benchmark was run using an NVIDIA RTX A6000, an AMD EPYC 7452 32-Core Processor and 528 Gigabyte RAM.

### E.2.1 `GASP!` vs SymPy

Another option is to compare the whole GASP! pipeline, including PA [Morettin et al., 2017] to enumerate the convex Polytopes, to a completely orthogonal approach. XADD [Kolb et al., 2018a] tackles the weighted model integral via symbolic manipulations. In practice, this means integrating via an explicit anti-derivative and then replacing the variable with the

symbolic lower and upper bounds, analogously on how solving an integral by hand is done. As the symbolic polynomial is represented using SymPy [Meurer et al., 2017b], our polynomial over both variables and coefficients can be naturally expressed and running XADD [Kolb et al., 2018a] on this polynomial directly results in $I_\phi(\mathbf{\Lambda})$. We benchmark GASP! vs. XADD equipped with SymPy on the $N$Star-constraints, which is just an $N$-pointed star where we always connect with opposite points. An example of the constraints can be seen in figure 9. Using our algorithm GASP!, we were able to reduce the runtime for the integral a polynomial of total degree 12 for a star with 17 corners from 6 hours and 20 minutes to 19 seconds, a significant speedup of 3 magnitudes or approximately 1200 times faster. Detailed results for the benchmark are in the Appendix in table 4 and 5.

| total degree
$N$ | 0 | 1 | 4 | 6 | 8 | 10 | 12 |
|---|---|---|---|---|---|---|---|
| 3 | 00:00:00 | 00:00:00 | 00:00:00 | 00:00:02 | 00:00:05 | 00:00:17 | 00:00:40 |
| 5 | 00:00:01 | 00:00:01 | 00:00:05 | 00:00:14 | 00:00:41 | 00:02:09 | 00:05:19 |
| 7 | 00:00:04 | 00:00:06 | 00:00:14 | 00:00:42 | 00:02:04 | 00:07:07 | 00:17:55 |
| 9 | 00:00:10 | 00:00:14 | 00:00:33 | 00:01:43 | 00:05:14 | 00:19:19 | 00:49:46 |
| 11 | 00:00:19 | 00:00:25 | 00:01:01 | 00:03:14 | 00:10:02 | 00:37:02 | 01:36:09 |
| 13 | 00:00:31 | 00:00:41 | 00:01:38 | 00:05:08 | 00:15:51 | 00:59:36 | 02:36:15 |
| 15 | 00:00:53 | 00:01:09 | 00:02:42 | 00:08:21 | 00:25:55 | 01:37:29 | 04:19:17 |
| 17 | 00:01:17 | 00:01:39 | 00:03:46 | 00:11:28 | 00:35:24 | 02:16:39 | 06:20:02 |

Table 4: We show the results for integrating the $N$Star-Benchmark using the XADD-Algorithm equipped with SymPy [Kolb et al., 2018a]. Results in $hh:mm:ss.$. This benchmark was run on the same machine as in 5. This benchmark was run using an NVIDIA RTX A6000, an AMD EPYC 7452 32-Core Processor and 528 Gigabyte RAM. The results for GASP! are provided in table 5.

| total degree
$N$ | 0 | 1 | 4 | 6 | 8 | 10 | 12 |
|---|---|---|---|---|---|---|---|
| 3 | 00:00:00 | 00:00:00 | 00:00:00 | 00:00:00 | 00:00:01 | 00:00:04 | 00:00:10 |
| 5 | 00:00:00 | 00:00:00 | 00:00:00 | 00:00:00 | 00:00:01 | 00:00:06 | 00:00:15 |
| 7 | 00:00:00 | 00:00:00 | 00:00:00 | 00:00:00 | 00:00:01 | 00:00:06 | 00:00:16 |
| 7 | 00:00:00 | 00:00:00 | 00:00:00 | 00:00:00 | 00:00:01 | 00:00:06 | 00:00:16 |
| 9 | 00:00:00 | 00:00:00 | 00:00:00 | 00:00:01 | 00:00:02 | 00:00:07 | 00:00:16 |
| 11 | 00:00:01 | 00:00:01 | 00:00:01 | 00:00:01 | 00:00:02 | 00:00:07 | 00:00:16 |
| 13 | 00:00:02 | 00:00:02 | 00:00:02 | 00:00:02 | 00:00:03 | 00:00:08 | 00:00:17 |
| 15 | 00:00:02 | 00:00:02 | 00:00:02 | 00:00:03 | 00:00:04 | 00:00:09 | 00:00:18 |
| 17 | 00:00:03 | 00:00:03 | 00:00:03 | 00:00:04 | 00:00:05 | 00:00:10 | 00:00:19 |

Table 5: GASP! is significantly faster compared to XADD [Kolb et al., 2018a]. We show results for integrating the $N$Star-Benchmark using GASP!. Results in $hh:mm:ss.$. This benchmark was run using an NVIDIA RTX A6000, an AMD EPYC 7452 32-Core Processor and 528 Gigabyte RAM. The results for XADD are provided in table 4.

### E.3 $N$STAR

For the $N$Star dataset, we generate $800.000$ $(\boldsymbol{x}, \boldsymbol{y})$-points on the $N$Star via rejection sampling. The $N$Star-constraints are formed by taking the $N$-pointed star on the circle with radius 10, where the the constraints are constructed by connecting each corner-point with the two most-opposite points. The distribution of $\mathbf{X}$ is uniform over the star, the distribution of $\mathbf{Y}$ is a cauchy-distribution with location $\mathbf{X}$ and scale $1.5\sqrt{10}$. The star is located in $[-10, 10]^2$.

We form train, test and validation datasets by dividing the points with a share of $70\%$, $15\%$ and $15\%$.

### E.3.1 Models

All models are trained with a batch-size of 512.

**PAL** We will now describe our settings for the `PAL`-models on the $N$Star-Dataset. For every variant of the star, we perform a grid-search over the following configurations, picking the best-performing according to the log-likelihood on the held-out dataset:

- epochs: 1500
- learning rate:$1e - 06$, $1e - 05$
- network hidden layers: $[1024, 1024]$, $[1024]$

The network is a fully-connected neural network using ReLU [Glorot et al., 2011] as activations. We use the schedule-free version of Adam [Defazio et al., 2024]. We use a single polynomials, with the re-parametrization as detailed in Appendix B.1 and train using a loss composed of log-likelihood and a penalty on very large integral-values detailed in Appendix B.3.

**GMM** For the GMM-Models, we perform the following grid-search:

- covariance: full and independent (although full covariance leads to better performing models for our dataset)
- epochs: 1500
- learning rate: $1e - 04$, $1e - 05$
- network hidden layers: $[1024, 1024]$, $[1024]$

We use the Adam optimizer [Kingma and Ba, 2015a].

**DSP** For DeepSeaProbLog, when performing the grid-search, we select the model according to the loss with the constant, final multiplier for the continous approximation for the inequality. The grid is over the following parameters:

- epoch: 1500
- learning rate: 0.001, 0.0001, 0.00001
- minimum-multiplier for the inequality-relaxation: 0.1, 1.0
- maximum-multiplier: 5
- network hidden layers: $[1024, 1024]$, $[1024]$
- optimiser: AdaMax and Adam [Kingma and Ba, 2015a]

Due to the slower training speed, we train DSP with a patience of 200-epochs (the other models are picked as the model with the best validation-score over all 1500 epochs). Finally, we train with sampling 50 times per input $\boldsymbol{x}^{(b)}$ in order to approximate $P(valid)$.

DSP is trained by optimizing both the fit on the data, as well as constraint satisfaction:

$$l_{dsp}(\boldsymbol{x}^{[1:b]}, \boldsymbol{y}^{[1:b]}) = \sum_i (-1) \cdot \log p(\boldsymbol{y}^i | \boldsymbol{x}^i) + \mathsf{CE}(p(\mathbf{Y} \models \phi \mid \mathbf{X} = \boldsymbol{x}^i)) \qquad \text{(DSP-Loss)}$$

where $p(\mathbf{Y} \models \phi \mid \mathbf{X} = \boldsymbol{x}^i)$ is approximated numerically in DSP in comparison to `PAL`. CE denotes the binary cross-entropy loss against the constant 1 label.

### E.3.2 Results

| N | NN + PAL | | | NN + GMM | | | | DeepSeaProblog | |
| --- | --- | --- | --- | --- | --- | --- | --- | --- | --- |
| | deg 10 | deg 14 | deg 18 | $K=1$ | $K=4$ | $K=8$ | $K=32$ | by LL | by Loss |
| 3 | -4.749 ±0.169 | **-4.674** ±**0.009** | -4.840 ±0.251 | -5.016 ±0.001 | -4.792 ±0.003 | -4.740 ±0.003 | -4.723 ±0.006 | -5.027 ±0.012 | -42.821 ±41.989 |
| 7 | -4.529 ±0.008 | **-4.527** ±**0.009** | -4.741 ±0.287 | -5.009 ±0.000 | -4.850 ±0.014 | -4.708 ±0.007 | -4.612 ±0.005 | -5.019 ±0.010 | -206.411 ±132.409 |
| 11 | **-4.570** ±**0.223** | -4.584 ±0.171 | -4.591 ±0.180 | -4.988 ±0.000 | -4.922 ±0.015 | -4.791 ±0.018 | -4.620 ±0.008 | -83.042 ±58.990 | -43.151 ±42.199 |
| 19 | -4.506 ±0.034 | **-4.492** ±**0.004** | 4.616 ±0.228 | -4.995 ±0.000 | -4.981 ±0.003 | -4.925 ±0.010 | -4.652 ±0.003 | -5.001 ±0.001 | -155.139 ±101.533 |

Table 6: Average log-likelihood on the test-set for the $N$Star-dataset. We test on a 3, 7, 11 and 19-Star and compare our approach (NN+PAL) to a conditional GMM and the neural distributional fact fitted by DeepSeaProblog. For DeepSeaProblog, we report the performance of two model, one selected by best log-likelihood (LL) and one by best loss (Loss), which takes into consideration both the fit and the probability of violating the constraints. After choosing the hyper-parameters, all runs were repeated 10-times and we report mean and standard deviation.

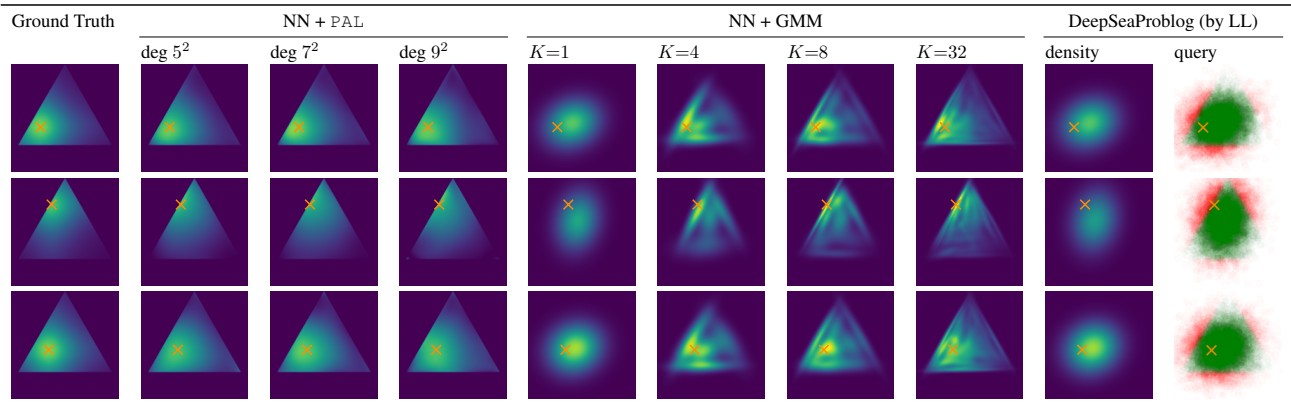

Table 7: Densities of the Ground-Truth compared to the polynomial, GMM and DeepSeaProbLog for the 3-Star problem with a cauchy-density. For the DSP model selected by log-likelihood, we show the density of the neural distributional fact, and we also show the result of querying the ProbLog program representing our constraints 10000 times. The samples associated with an true-label are shown in green, the samples associated with a false-label are shown in red.

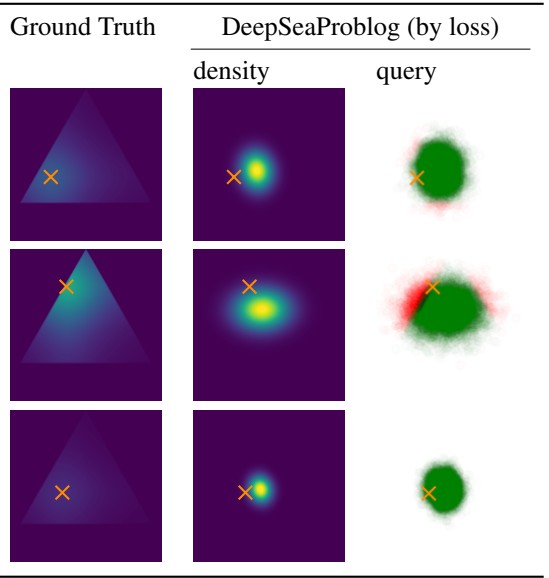

Table 8: Densities of the Ground-Truth compared to the DeepSeaProbLog for the 3Star problem with a cauchy-density. DeepSeaProbLog is selected by loss, and due to avoiding the constraints, more concentrated and therefore visualized separately. We show the density for the neural distributional fact and the samples obtained by the query. The samples associated with an true-label are shown in green, the samples associated with a false-label are shown in red. We choose to visualize this separately in order to keep the color-scheme in figure 7 reasonable.

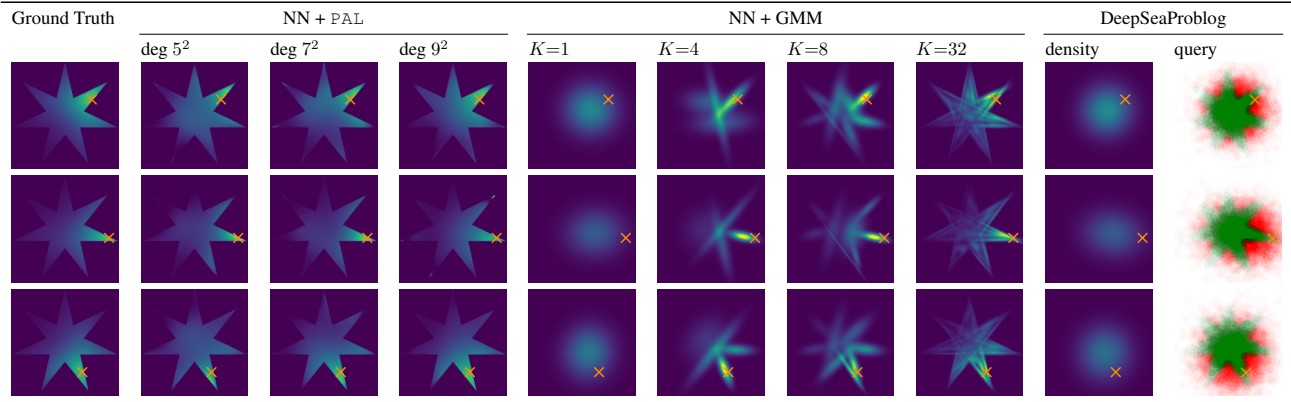

Table 9: Densities of the Ground-Truth compared to the polynomial, GMM and DeepSeaProbLog for the 7-Star problem with a cauchy-density. For the DSP model selected by log-likelihood, we show the density of the neural distributional fact, and we also show the result of querying the ProbLog program representing our constraints 10000 times. The samples associated with an true-label are shown in green, the samples associated with a false-label are shown in red.

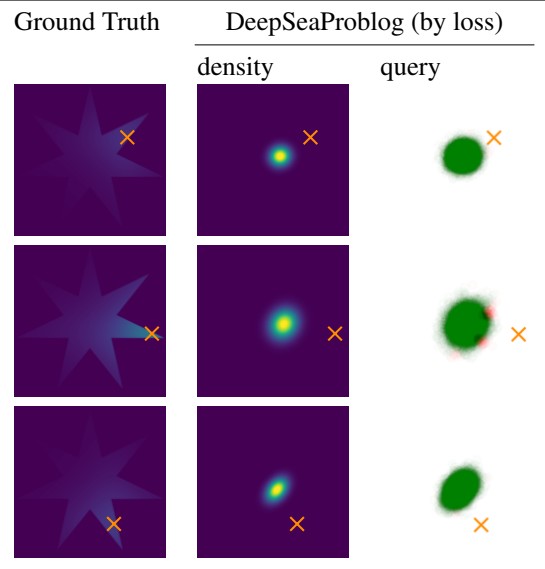

Table 10: Densities of the Ground-Truth compared to the DeepSeaProbLog for the 7Star problem with a cauchy-density. DeepSeaProbLog is selected by loss, and due to avoiding the constraints, more concentrated and therefore visualized separately. We show the density for the neural distributional fact and the samples obtained by the query. The samples associated with an true-label are shown in green, the samples associated with a false-label are shown in red. We choose to visualize this separately in order to keep the color-scheme in figure 9 reasonable.

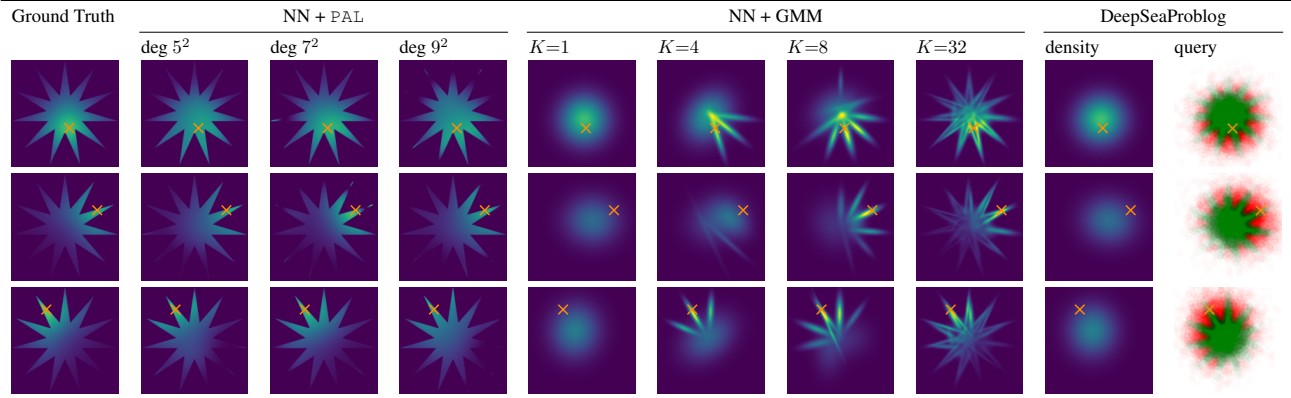

Table 11: Densities of the Ground-Truth compared to the polynomial, GMM and DeepSeaProbLog for the 11-Star problem with a cauchy-density. For the DSP model selected by log-likelihood, we show the density of the neural distributional fact, and we also show the result of querying the ProbLog program representing our constraints 10000 times. The samples associated with an true-label are shown in green, the samples associated with a false-label are shown in red.

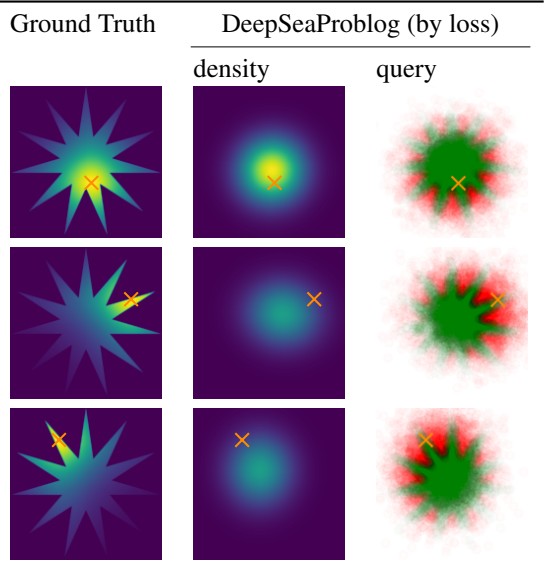

Table 12: Densities of the Ground-Truth compared to the DeepSeaProbLog for the 11Star problem with a cauchy-density. DeepSeaProbLog is selected by loss, and due to avoiding the constraints, more concentrated and therefore visualized separately. We show the density for the neural distributional fact and the samples obtained by the query. The samples associated with an true-label are shown in green, the samples associated with a false-label are shown in red. We choose to visualize this separately in order to keep the color-scheme in figure 11 reasonable.

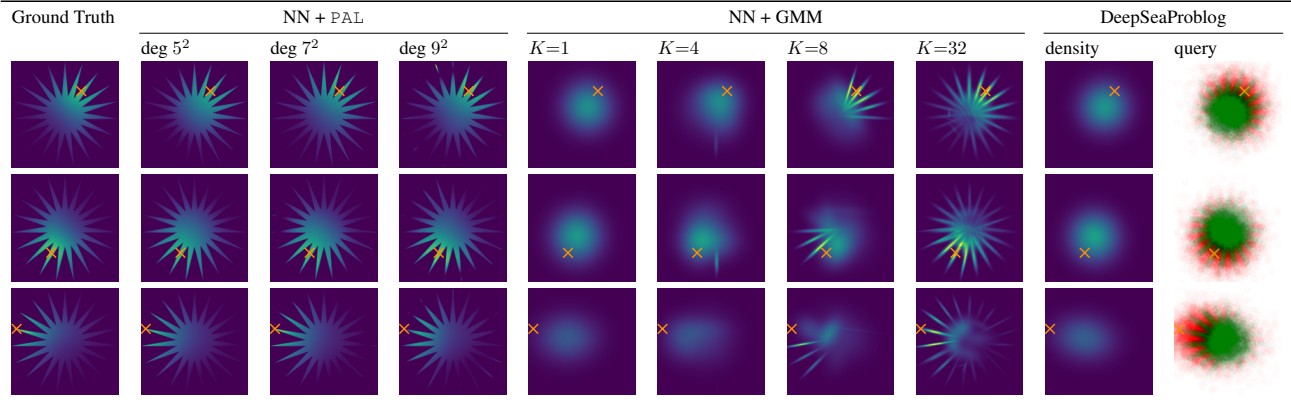

Table 13: Densities of the Ground-Truth compared to the polynomial, GMM and DeepSeaProbLog for the 19-Star problem with a cauchy-density. For the DSP model selected by log-likelihood, we show the density of the neural distributional fact, and we also show the result of querying the ProbLog program representing our constraints 10000 times. The samples associated with a true-label are shown in green, the samples associated with a false-label are shown in red.

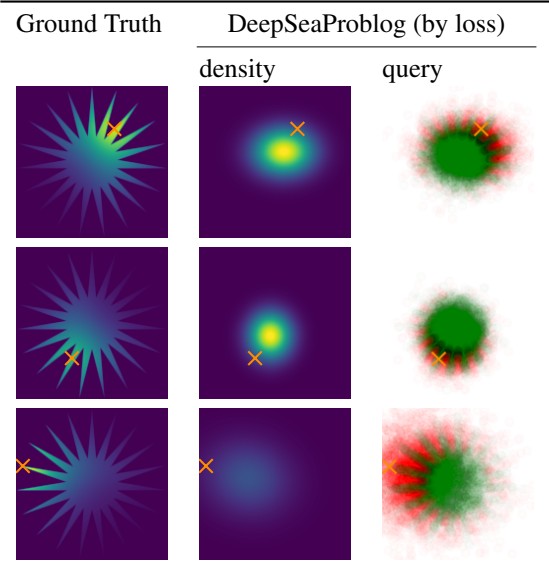

Table 14: Densities of the Ground-Truth compared to the DeepSeaProbLog for the 19Star problem with a cauchy-density. DeepSeaProbLog is selected by loss, and due to avoiding the constraints, more concentrated and therefore visualized separately. We show the density for the neural distributional fact and the samples obtained by the query. The samples associated with an true-label are shown in green, the samples associated with a false-label are shown in red. We choose to visualize this separately in order to keep the color-scheme in figure 13 reasonable.

### E.4 STANFORD DRONE DATASET - JOINT DISTRIBUTION

#### E.4.1 Details of the Dataset - Scenario 1

We focus on image 12, the image with the most trajectories. We first want to note that the time-resolution, due to it being extracted from a 30-fps video, is quite high. In order to eliminate outliers, we first delete all points without movement (trajectories with a total variance of less than 20 and points with a distance of less than 0.1 compared to the previous). Then, we clean the data by discarding all short trajectories (= length of less than 50). We arrive at 415 moving trajectories out of the 499 we started with. We split this dataset, by trajectory, into train, test and validation (70%, 15% and 15%) and then concatenate all the points. We arrive at 124998 train points, 26786 validation points and 26786 test points. While this appears like a significant of points, we want to stress that many are heavily autocorrelated due to the high resolution, and the actual, total, amount of trajectories is only 415.

| type | detail | num. params | $\Pr(\neg\phi)$ | log-like |
|------|--------|------------|-----------|----------|
| PAL (Spline) | 8 knots, 10 mixtures | 330 | **0.000** | -3.013 $\pm$0.002 |
| PAL (Spline) | 8 knots, 8 mixtures | 264 | **0.000** | 3.023 $\pm$0.003 |
| PAL (Spline) | 8 knots, 4 mixtures | 132 | **0.000** | -3.067 $\pm$0.009 |
| PAL (Spline) | 10 knots, 10 mixtures | 410 | **0.000** | -2.984 $\pm$0.002 |
| PAL (Spline) | 10 knots, 8 mixtures | 328 | **0.000** | -2.995 $\pm$0.005 |
| PAL (Spline) | 10 knots, 4 mixtures | 164 | **0.000** | -3.045 $\pm$0.008 |
| PAL (Spline) | 12 knots, 10 mixtures | 490 | **0.000** | -2.971 $\pm$0.003 |
| PAL (Spline) | 12 knots, 8 mixtures | 392 | **0.000** | -2.979 $\pm$0.003 |
| PAL (Spline) | 12 knots, 4 mixtures | 196 | **0.000** | -3.025 $\pm$0.010 |
| PAL (Spline) | 14 knots, 10 mixtures | 570 | **0.000** | -2.950 $\pm$0.002 |
| PAL (Spline) | 14 knots, 8 mixtures | 456 | **0.000** | -2.961 $\pm$0.002 |
| PAL (Spline) | 14 knots, 4 mixtures | 228 | **0.000** | -3.009 $\pm$0.008 |
| PAL (Spline) | 16 knots, 10 mixtures | 650 | **0.000** | -2.937 $\pm$0.003 |
| PAL (Spline) | 16 knots, 8 mixtures | 520 | **0.000** | -2.948 $\pm$0.003 |
| PAL (Spline) | 16 knots, 4 mixtures | 260 | **0.000** | -2.998 $\pm$0.010 |
| GMM | $K=5$ | 30 | $\approx$ 12.475 $\pm$0.761 | -3.359 $\pm$0.034 |
| GMM | $K=10$ | 60 | $\approx$ 8.887 $\pm$0.224 | -3.223 $\pm$0.008 |
| GMM | $K=20$ | 120 | $\approx$ 4.229 $\pm$0.142 | -3.081 $\pm$0.012 |
| GMM | $K=50$ | 300 | $\approx$ 2.375 $\pm$0.112 | -2.983 $\pm$0.004 |
| GMM | $K=100$ | 600 | $\approx$ 1.190 $\pm$0.052 | **-2.917 $\pm$0.005** |
| Flow | 1 transformation ($t$), 128x2 hidden | 22830 | $\approx$ 5.643 $\pm$0.487 | -3.098 $\pm$0.013 |
| Flow | 1 transformation ($t$), 64x2 hidden | 7342 | $\approx$ 5.245 $\pm$0.516 | -3.089 $\pm$0.017 |
| Flow | 1 transformation ($t$), 32x2 hidden | 2670 | $\approx$ 5.651 $\pm$0.596 | -3.109 $\pm$0.016 |
| Flow | 2 transformations ($t$), 128x2 hidden | 45660 | $\approx$ 2.616 $\pm$0.221 | -2.986 $\pm$0.008 |
| Flow | 2 transformations ($t$), 64x2 hidden | 14684 | $\approx$ 2.157 $\pm$0.250 | -2.972 $\pm$0.011 |
| Flow | 2 transformations ($t$), 32x2 hidden | 5340 | $\approx$ 2.468 $\pm$0.612 | -2.979 $\pm$0.016 |
| Flow | 5 transformations ($t$), 128x2 hidden | 114150 | $\approx$ 1.771 $\pm$0.159 | -2.940 $\pm$0.007 |
| Flow | 5 transformations ($t$), 64x2 hidden | 36710 | $\approx$ 1.930 $\pm$0.130 | -2.949 $\pm$0.007 |
| Flow | 5 transformations ($t$), 32x2 hidden | 13350 | $\approx$ 1.677 $\pm$0.231 | -2.943 $\pm$0.012 |
| Flow | 10 transformations ($t$), 128x2 hidden | 228300 | $\approx$ 1.502 $\pm$0.109 | -2.919 $\pm$0.007 |
| Flow | 10 transformations ($t$), 64x2 hidden | 73420 | $\approx$ 1.698 $\pm$0.202 | -2.943 $\pm$0.018 |
| Flow | 10 transformations ($t$), 32x2 hidden | 26700 | $\approx$ 1.826 $\pm$0.252 | -2.938 $\pm$0.007 |

Table 15: Results for the $p(\mathbf{Y})$-case of the Stanford-Drone dataset for scenario 1. All Spline models have equal number of knots in $y_1$ and $y_2$. The average percent of probability mass covering invalid space ($\Pr(\neg\phi)$) over our test-set is given in percent. After choosing the hyper-parameters, all runs were repeated 10-times and we report mean and standard deviation.

Before Filtering: After Filtering:

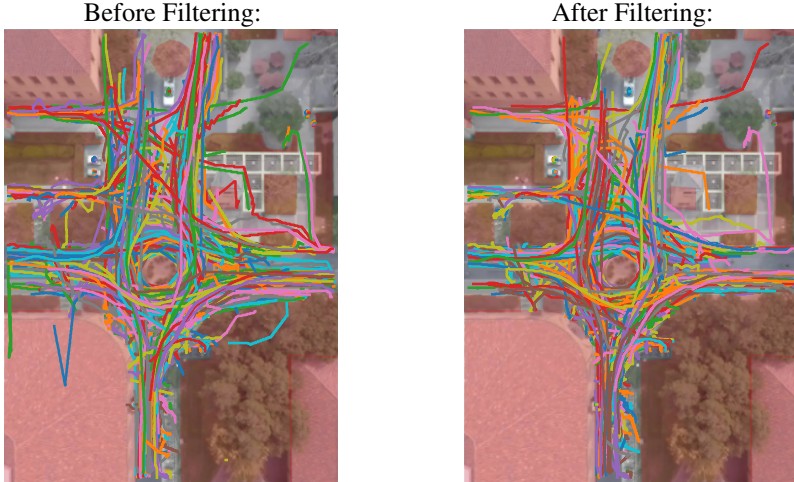

Figure 12: The trajectories before/after filtering for image 12 in the Stanford Drone Dataset. The constraints are shown in red.

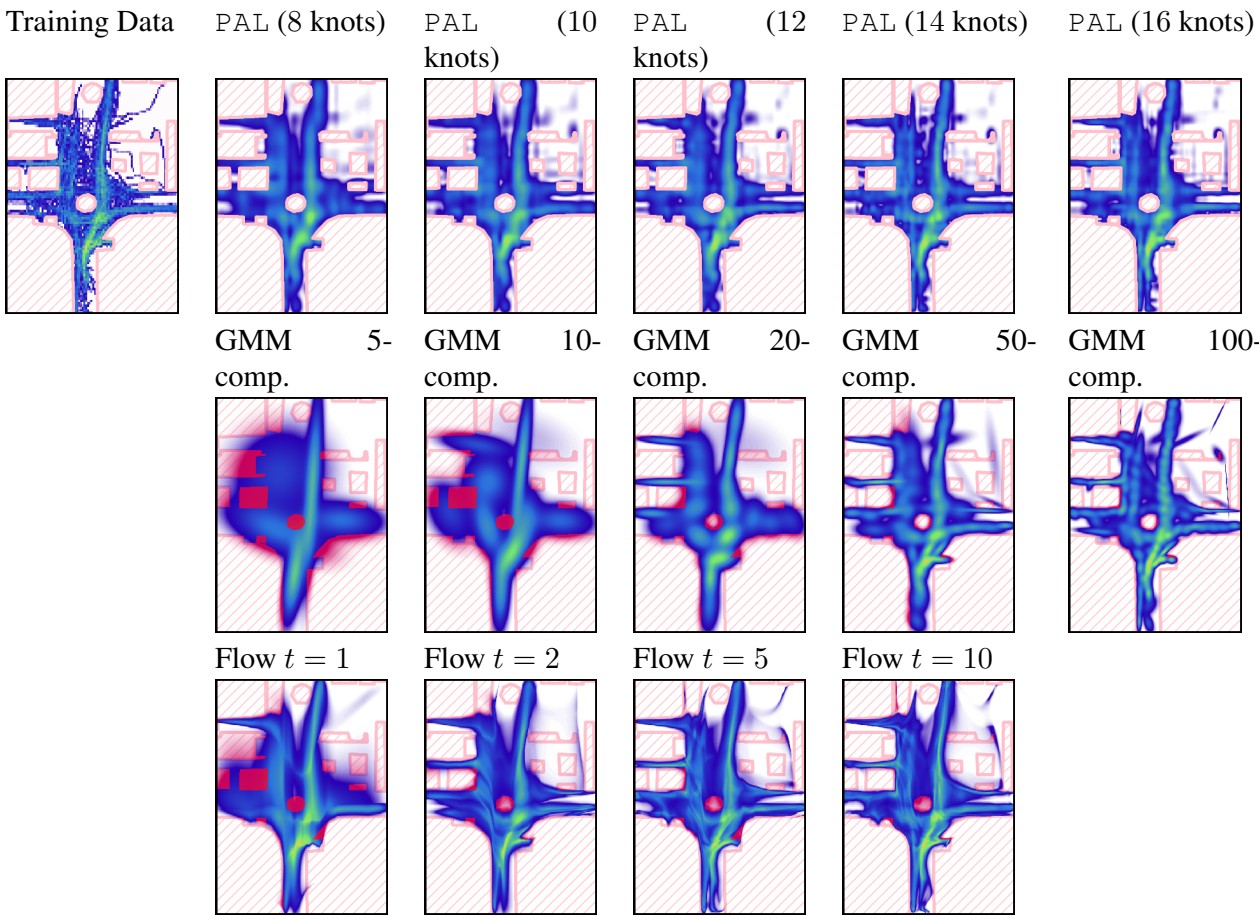

Table 16: Densities on the $p(\mathbf{Y})$-case of the Stanford-Drone dataset for scenario 1. All Spline models have 10-mixture components and equal number of knots in $Y_1$ and $Y_2$. The flow-models have two hidden layers of size 128 per transformation.

### E.4.2 Details of the Dataset - Scenario 2

We focus on image 2, which consists only of 119 trajectories. We perform the same filtering etc. as in scenario 1 and arrive at 53504 train points, 10197 validation points and 13860 test points.

| type | detail | num. params | perc. invalid | log-like |
|------|--------|-------------|---------------|----------|
| PAL (Spline) | 8 knots, 10 mixtures | 330 | **0.000** | -3.376 ±0.007 |
| PAL (Spline) | 8 knots, 8 mixtures | 264 | **0.000** | -3.376 ±0.007 |
| PAL (Spline) | 10 knots, 10 mixtures | 410 | **0.000** | -3.348 ±0.005 |
| PAL (Spline) | 10 knots, 8 mixtures | 328 | **0.000** | -3.362 ±0.006 |
| PAL (Spline) | 12 knots, 10 mixtures | 490 | **0.000** | -3.329 ±0.003 |
| PAL (Spline) | 12 knots, 8 mixtures | 392 | **0.000** | -3.343 ±0.004 |
| PAL (Spline) | 14 knots, 10 mixtures | 570 | **0.000** | -3.313 ±0.003 |
| PAL (Spline) | 14 knots, 8 mixtures | 456 | **0.000** | -3.333 ±0.007 |
| PAL (Spline) | 16 knots, 10 mixtures | 650 | **0.000** | -3.301 ±0.004 |
| PAL (Spline) | 16 knots, 8 mixtures | 520 | **0.000** | -3.322 ±0.004 |
| GMM | $K=5$ | 35 | ≈ 12.220 ±0.053 | -3.701 ±0.002 |
| GMM | $K=10$ | 70 | ≈ 6.589 ±0.283 | -3.564 ±0.012 |
| GMM | $K=20$ | 140 | ≈3.223 ±0.542 | -3.449 ±0.019 |
| GMM | $K=50$ | 350 | ≈ 1.289 ±0.159 | -3.351 ±0.014 |
| GMM | $K=100$ | 700 | ≈ 0.656 ±0.042 | **-3.259 ±0.005** |
| Flow | 1 transformation ($t$), 32x2 hidden | 2670 | ≈ 2.274 ±0.214 | -3.431 ±0.012 |
| Flow | 2 transformations ($t$), 32x2 hidden | 5340 | ≈ 1.210 ±0.285 | -3.332 ±0.020 |
| Flow | 5 transformations ($t$), 32x2 hidden | 13350 | ≈ 0.710 ±0.126 | -3.266 ±0.015 |
| Flow | 10 transformations ($t$), 32x2 hidden | 26700 | ≈ 0.867 ±0.126 | -3.273 ±0.012 |

Table 17: Results for the $p(\mathbf{Y})$-case of the Stanford-Drone dataset for scenario 2. All Spline models have equal number of knots in $y_1$ and $y_2$. The average percent of probability mass covering invalid space ($\Pr(\neg\phi)$) over our test-set is given in percent. After choosing the hyper-parameters, all runs were repeated 10-times and we report mean and standard deviation.

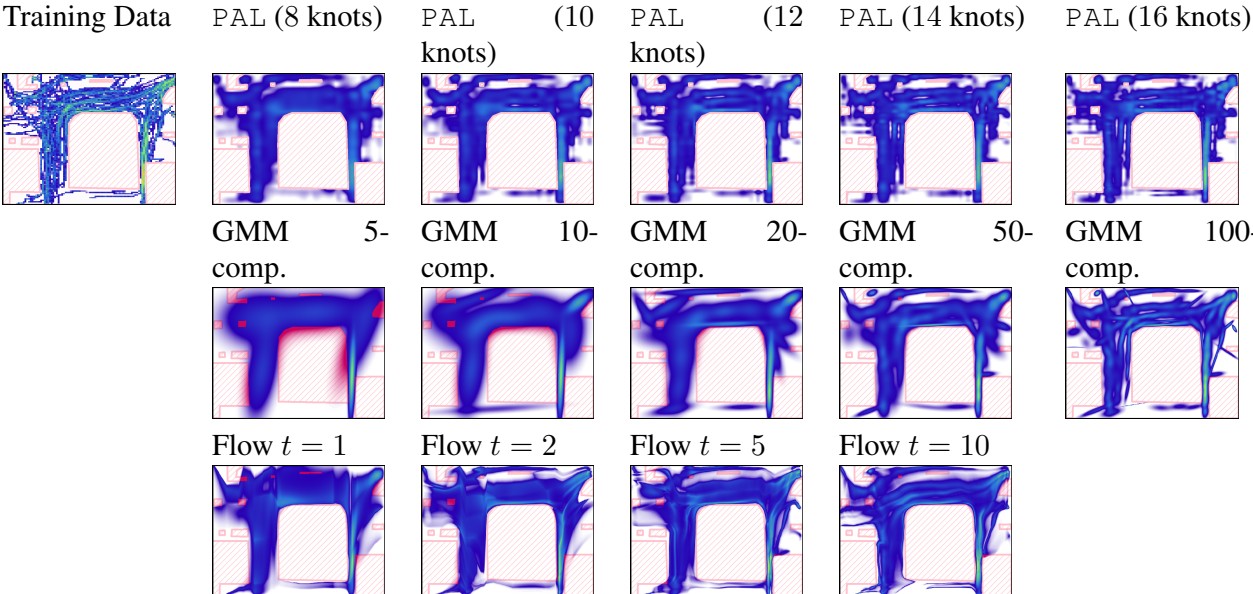

Table 18: Densities on the $p(\mathbf{Y})$-case of the Stanford-Drone dataset for scenario 2. All Spline models have 10-mixture components and equal number of knots in $Y_1$ and $Y_2$. The flow-models have two hidden layers of size $32x2$ per transformation.

### E.5 STANFORD DRONE DATASET - CONDITIONAL DISTRIBUTIONS

#### E.5.1 Details of the Dataset

The goal in this task is to fit the distribution of possible future trajectories, so if our random variable for the coordinates at timestep $t$ is $\mathbf{C}_t$, then our goal is to predict $\mathbf{Y} = \mathbf{C}_{t \geq t'}$ given the current a window of 5 equidistant points $\mathbf{X} = (\mathbf{C}_{t=t'-0\cdot\Delta}, \mathbf{C}_{t=t'-1\cdot\Delta}, \dots, \mathbf{C}_{t=t'-5\cdot\Delta})$, with $\Delta$ being some step-size. This challenging construction induces multi-modality and uncertainty in the predictive distribution, because if there is a chance of visiting a certain area in the future given the 5 steps, it must have some probability mass assigned to it.

In order to bias our models towards more well-connected paths, we bias the network towards the closer in time datapoints. The distribution is therefore a mixture of both a uniform distribution over the whole future trajectory and a uniform distribution over the future trajectory of length step-size $s$, so 70 in our case. Both choices have equal chance.

#### E.5.2 Scenario 1

For this task, we focus on image 12, the image with the most trajectories. We want to note that the time-resolution, due to it being extracted from a 30-fps video, is quite high. As we want to focus on the trajectory, we eliminate all points without movement, so points with a distance of less than $0.1$ compared to the previous. We take a step-size of 70 for the window of 5 points that form our $\mathbf{X}$, which we will slide through the trajectory. We also discard all trajectories that are too short to fill our window, so where the length is less than $5 \cdot 70$, as we want to focus on long trajectories. We split the trajectories into 70% train, 15% validation and 15% test. We then create a static validation and test-dataset by sampling 10 future $\mathbf{Y}$ points per window $\mathbf{X} = x$ at creation statically. For the train-dataset, we sample the during training, so for the same $\mathbf{X} = x$ it will see different future points. We arrive at 22619-train datapoints, with $\mathbf{Y}$ dynamically sampled per $\mathbf{X} = x$, 42740-validation datapoints and 49660-test datapoints.

#### Model Details

All our models are simple, fully connected neural networks with ReLu as an activation function [Glorot et al., 2011].

We denote the following sizes:

- **large**: 2 hidden layers of size 2048 each
- **medium**: 2 hidden layers of size 1024 each
- **small**: 2 hidden layers of size 512 each

We train all models for a maximum of 500 epochs with a patience of 20 epochs and run the hyper-parameter search for each network-size per model-type.

`PAL` For the `PAL` models, we do a grid-search over the following parameters:

- **number of mixtures**: 8 and 10
- **number of knots**: 10 and 14 (equal over $y_1$ and $y_2$)
- **optimizer** AdamW schedulefree [Defazio et al., 2024] with learning rate: 0.001, 0.0001, 0.00001 and batch-sizes 16, 32 and 128
- **net-sizes**: large/medium/small

**GMM** For the conditional GMM-models, we do a grid-search over the following parameters:

- **number of components** $K$: 4, 32, 50, 80, 100 with full covariances
- **net-sizes**: large/medium/small
- **optimizer** Adam [Kingma and Ba, 2015b] with learning rates: 0.001, 0.0001, 0.00001 and batch-sizes 16, 32 (128 led to worse performance due to overfitting on initial-runs and was excluded)

**DSP** For the DSP models, we do a grid-search over the following parameters:

- **optimizer** AdaMax [Kingma and Ba, 2015b] with learning rates 0.01, 0.001, 0.0001, 0.00001 and batch-sizes 16, 32 (128 led to worse performance due to overfitting on initial-runs and was excluded)
- **annealing starting-multiplier**: 0.1, 1.0
- **end-multiplier**: 5
- **net-sizes**: large/medium/small

We use a tanh-scaling of the annealing multiplier with an alpha of $1e-4$ and train with the loss DSP-Loss.

**Results**

| type | size dist. | net size | log-like | num. params dist. | $\Pr(\neg\phi)$ |
|---|---|---|---|---|---|
| PAL (Spline) | 14 knots, 10 mixtures | medium | -2.209 ±0.136 | 570 | **0.000** |
| PAL (Spline) | 10 knots, 8 mixtures | medium | -2.942 ±1.204 | 328 | **0.000** |
| PAL (Spline) | 14 knots, 8 mixtures | small | **-2.086 ±0.144** | 456 | **0.000** |
| PAL (Spline) | 10 knots, 8 mixtures | small | -2.272 ±0.130 | 328 | **0.000** |
| PAL (Spline) | 14 knots, 10 mixtures | large | -2.174 ±0.120 | 570 | **0.000** |
| PAL (Spline) | 10 knots, 8 mixtures | large | -2.263 ±0.168 | 328 | **0.000** |
| GMM | $K$=100 | medium | -3.323 ±0.738 | 700 | ≈ 20.191 ±1.937 |
| GMM | $K$=80 | medium | -11.173 ±8.202 | 560 | ≈ 72.969 ±34.192 |
| GMM | $K$=50 | medium | -2.932 ±0.361 | 350 | ≈ 21.706 ±2.727 |
| GMM | $K$=32 | medium | -2.989 ±0.345 | 224 | ≈ 21.620 ±2.672 |
| GMM | $K$=4 | medium | -3.262 ±0.379 | 28 | ≈ 20.955 ±1.366 |
| GMM | $K$=100 | small | -2.838 ±0.485 | 700 | ≈ 20.321 ±1.672 |
| GMM | $K$=80 | small | -2.835 ±0.410 | 560 | ≈ 20.096 ±1.964 |
| GMM | $K$=50 | small | -2.644 ±0.258 | 350 | ≈ 20.989 ±1.774 |
| GMM | $K$=32 | small | -2.735 ±0.379 | 224 | ≈ 19.842 ±1.584 |
| GMM | $K$=4 | small | -3.235 ±0.970 | 28 | ≈ 21.859 ±1.600 |
| GMM | $K$=50 | large | -3.074 ±0.273 | 350 | ≈ 24.721 ±3.632 |
| GMM | $K$=32 | large | -6.383 ±6.179 | 224 | ≈ 36.660 ±33.943 |
| GMM | $K$=100 | large | -4.963 ±4.690 | 700 | ≈ 30.516 ±24.794 |
| GMM | $K$=80 | large | -16.420 ±9.413 | 560 | ≈ 83.409 ±31.042 |
| GMM | $K$=4 | large | -7.553 ±5.529 | 28 | ≈ 48.345 ±35.762 |
| DSP (by loss) | 1 Gaussian | large | -8.532 ±11.325 | 6 | ≈ 29.035 ±3.074 |
| DSP (by loss) | 1 Gaussian | medium | -6.176 ±5.132 | 6 | ≈ 36.648 ±9.670 |
| DSP (by loss) | 1 Gaussian | small | -3.876 ±0.466 | 6 | ≈ 49.046 ±16.395 |
| DSP (by log-like) | 1 Gaussian | large | -8.520 ±11.330 | 6 | ≈ 34.963 ±9.933 |
| DSP (by log-like) | 1 Gaussian | medium | -6.152 ±5.144 | 6 | ≈ 33.322 ±5.761 |
| DSP (by log-like) | 1 Gaussian | small | -3.861 ±0.480 | 6 | ≈ 58.209 ±15.949 |

Table 19: Results for the $p(\mathbf{Y}|\mathbf{X})$-case of the Stanford-Drone dataset for scenario 1. All Spline models have equal number of knots in $y_1$ and $y_2$. The average percent of probability mass covering invalid space ($\Pr(\neg\phi)$) over our test-set is given in percent. It is approximated by sampling $10^6$ times per datapoint $\boldsymbol{x}$ in the test-set, computing constraint satisfaction, and then taking the average. After choosing the hyper-parameters, all runs were repeated 10-times and we report mean and standard deviation.

Table 20: Densities for the predictive positions for the $P(\mathbf{Y} \mid \mathbf{X})$ case on the stanford drone dataset for scenario 1. We compare the best $4$ spline-models against the best $4$ GMM models and the best DSP models both by log-likelihood and loss from 19. The colormap is normalized per sample.

| Model | Sample 1 | Sample 2 | Sample 3 | Sample 4 | Sample 5 | Sample 6 |
|---|---|---|---|---|---|---|

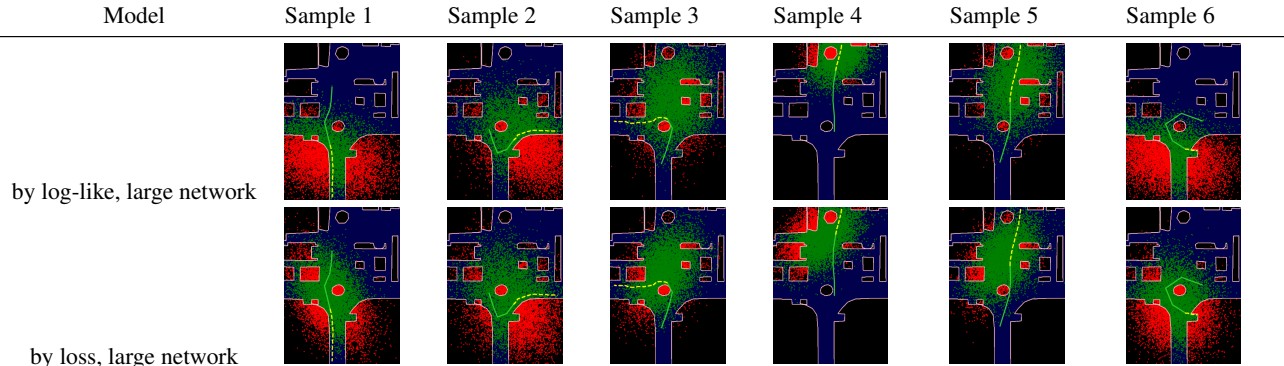

by log-like, large network

by loss, large network

Table 21: Samples from the Problog-Query representing the Constraints on the Stanford-Drone Dataset. We display the samples and the associated labels that we obtain from DeepSeaProbLog. We show the same samples as displayed in 20.

### E.5.3 Scenario 2

We provide further provide results on Image 2 from the Stanford Drone Dataset [Robicquet et al., 2016]. This is a small scenario, with 119 trajectories to start. For the conditional trajectory-prediction problem, we use the same setup is detailed in section E.5 and arrive at 21273 train, 72710 validation, and 62080 test-datapoints.

### Model Details

We narrow down our search-space and pick the best-performing configurations from E.5.2 to apply our grid-search on, but discard the large net-size as the dataset is smaller.

**PAL** For the PAL models, we do a grid-search over the following parameters:

- **number of mixtures**: 8 and 10
- **number of knots**: 10 and 14 (equal over $y_1$ and $y_2$)
- **optimizer** AdamW schedulefree [Defazio et al., 2024] with learning rate: 0.001, 0.0001 and batch-sizes 16, 32 and 128
- **net-sizes**: medium/small

**GMM** For the conditional GMM-models, we do a grid-search over the following parameters:

- **number of components** $K$: 4, 32, 50, 100 with full covariances
- **net-sizes**: medium/small
- **optimizer** Adam [Kingma and Ba, 2015b] with learning rates: 0.001, 0.0001 and batch-sizes 16, 32, 128

**DSP** For the DSP models, we do a grid-search over the following parameters:

- **optimizer** AdaMax [Kingma and Ba, 2015b] with learning rates 0.001, 0.0001 and batch-sizes 16, 32, 128
- **annealing starting-multiplier**: 0.1, 1.0
- **end-multiplier**: 5
- **net-sizes**: medium/small

We use a tanh-scaling of the annealing multiplier with an alpha of $1e - 4$ and train with the loss DSP-Loss.

### Results

| Image | Trajectories | Constraints |
|:-:|:-:|:-:|
| 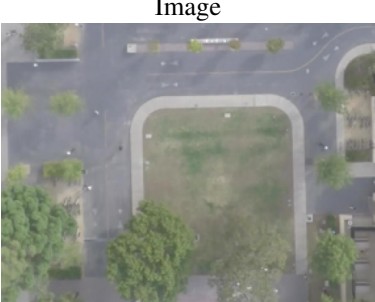 | 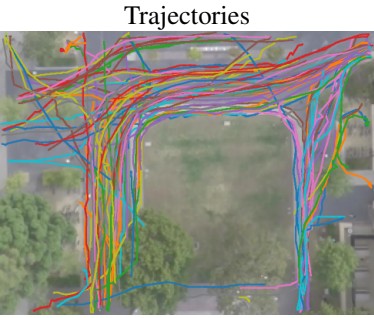 | 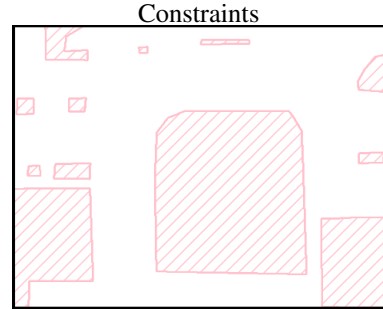 |

Figure 13: **Our dataset combines challenging constraints with real-world data** on trajectories (middle) and aerial maps (left) taken from Robicquet et al. [2016]. We manually label the data, indicating invalid areas to move to. This is image 2 from the Stanford drone dataset.

| type | size dist. | net size | log-like | num. params dist. | Pr($\neg\phi$) |
|------|-----------|----------|---------:|------------------:|---------------:|
| NN + PAL | 14 knots, 10 mixtures | medium | -2.237 ±0.268 | 570 | **0.000** |
| NN + PAL | 10 knots, 8 mixtures | medium | -2.302 ±0.175 | 328 | **0.000** |
| NN + PAL | 14 knots, 8 mixtures | small | -2.232 ±0.275 | 456 | **0.000** |
| NN + PAL | 10 knots, 10 mixtures | small | **-2.091 ±0.086** | 410 | **0.000** |
| NN + GMM | $K$=100 | medium | -2.816 ±0.205 | 700 | $\approx$ 15.435 ±4.099 |
| NN + GMM | $K$=50 | medium | -3.129 ±0.572 | 350 | $\approx$ 14.905 ±3.603 |
| NN + GMM | $K$=32 | medium | -2.714 ±0.228 | 224 | $\approx$ 16.399 ±3.852 |
| NN + GMM | $K$=4 | medium | -2.780 ±0.316 | 28 | $\approx$ 17.271 ±4.785 |
| NN + GMM | $K$=100 | small | -2.423 ±0.193 | 700 | $\approx$ 14.684 ±2.973 |
| NN + GMM | $K$=50 | small | -2.396 ±0.231 | 350 | $\approx$ 15.610 ±3.715 |
| NN + GMM | $K$=32 | small | -2.674 ±0.310 | 224 | $\approx$ 15.390 ±4.273 |
| NN + GMM | $K$=4 | small | -2.650 ±0.281 | 28 | $\approx$ 19.093 ±6.341 |
| DSP by loss | 1 Gaussian | medium | -3.611 ±0.287 | 6 | $\approx$ 35.986 ±6.946 |
| DSP by loss | 1 Gaussian | small | -30.006 ±76.260 | 6 | $\approx$ 52.929 ±17.107 |
| DSP by log-like | 1 Gaussian | medium | -3.611 ±0.287 | 6 | $\approx$ 36.148 ±1.641 |
| DSP by log-like | 1 Gaussian | small | -29.967 ±76.274 | 6 | $\approx$ 36.433 ±2.976 |

Table 22: Results for the $P(Y|X)$-case of the Stanford-Drone dataset for scenario 2. All Spline models have equal number of knots in $y_1$ and $y_2$. The average percent of probability mass covering invalid space ($\Pr(\neg\phi)$) over our test-set is given in percent. It is approximated by sampling $10^6$ times per datapoint $x$ in the test-set, computing constraint satisfaction, and then taking the average. After choosing the hyper-parameters, all runs were repeated 10-times and we report mean and standard deviation.

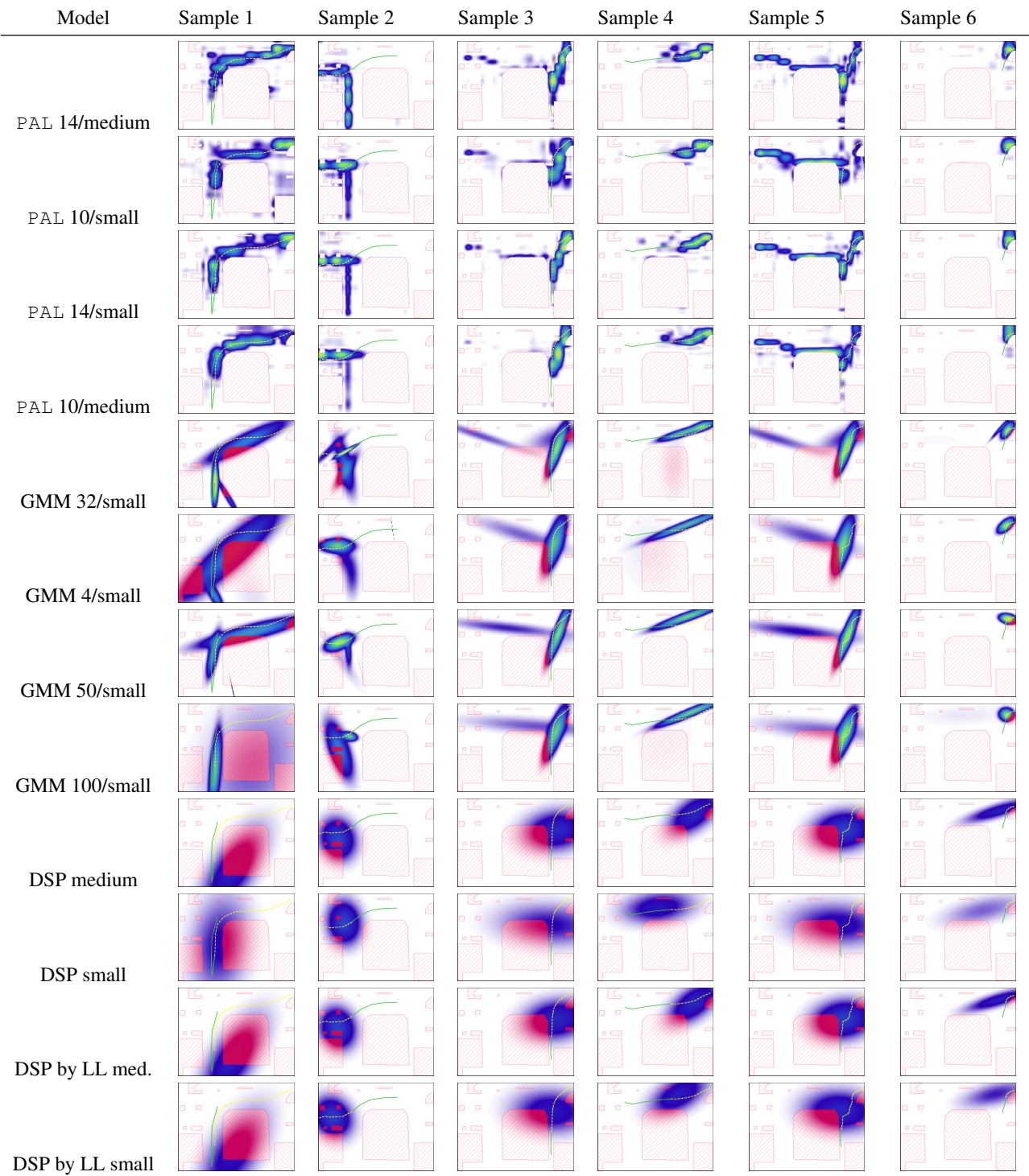

Table 23: Densities for the predictive positions for the $P(\mathbf{Y} \mid \mathbf{X})$ case on the Stanford drone dataset on scenario 2. We compare the best 4 spline models against the best 4 GMM models and the best DSP models both by log-likelihood and loss from 22. Colormap is normalized per sample.

