# OpenReview forum: "A Probabilistic Neuro-symbolic Layer for Algebraic Constraint Satisfaction"
_auai.org/UAI/2025/Workshop/TPM — TPM 2025_

### Official Review · Reviewer_KHqT · 2025-06-13
**Nice and mature paper, tractability is not central**

**Rating:** 3

**Review:**

This is a nice paper about embedding deterministic constraints as algebraic layers to be added at the bottom of a neural model. Earlier works (sometimes based on probabilistic circuits) have been proposed for discrete variables. Instead, this paper contains a dedicated and highly parallelisable approach for continuous variables that outperforms other approaches.

The paper is very mature, and, considering the number of pages, I think it is a "recently appeared paper" submission. I think it should be definitely accepted. Yet, I am not recommending it as a best paper for the only reason that the "tractability" aspects are not so relevant for the work.

---

### Official Review · Reviewer_dX4f · 2025-06-13
**Exact and GPU-efficient neuro-symbolic layer for enforcing algebraic constraints in continuous domains, achieving tractable inference, better accuracy**

**Rating:** 3

**Review:**

This work  presents a significant advancement in nesy models by introducing Probabilistic Algebraic Layer (PAL) a differentiable probabilistic layer that ensures exact satisfaction of complex, non-convex algebraic constraints over continuous variables. By combining symbolic reasoning with tractable probabilistic inference, PAL enables neural networks to produce zero-probability predictions for invalid configurations, a crucial feature for safety-critical applications.
The authors propose GASP!, a novel GPU-accelerated symbolic integration scheme that amortizes the cost of computing the normalization constant required for exact maximum likelihood training, outperforming existing solvers.
Empirical results across synthetic tasks and real-world trajectory modeling demonstrate PAL's ability to tightly respect hard constraints without sacrificing modeling flexibility or predictive accuracy something other baselines don't achieve. The paper is technically sound,  and well-written. I suggest to include uncertainty quantification, extending comparisons probabilistic programming methods, and clarifying any training-time trade-offs. I would also suggest to include a limitations and future work section for more clarity.